# LC/MS-MS Analysis of Phenolic Compounds in *Hyoscyamus albus* L. Extract: In Vitro Antidiabetic Activity, In Silico Molecular Docking, and In Vivo Investigation against STZ-Induced Diabetic Mice

**DOI:** 10.3390/ph16071015

**Published:** 2023-07-18

**Authors:** Sabrina Lekmine, Ouided Benslama, Kenza Kadi, Antonio Ignacio Martín-García, Mustafa Abdullah Yilmaz, Salah Akkal, Ali Boumegoura, Abdullah S. Alhomida, Mohammad Shamsul Ola, Ahmad Ali

**Affiliations:** 1Biotechnology, Water, Environment and Health Laboratory, Abbes Laghrour University, Khenchela 40000, Algeria; 2Laboratory of Natural Substances, Biomolecules, and Biotechnological Applications, Department of Natural and Life Sciences, Larbi Ben M’Hidi University, Oum El Bouaghi 04000, Algeria; 3Estación Experimental del Zaidín (CSIC) Profesor Albareda 1, 18008 Granada, Spain; 4Faculty of Pharmacy, Department of Analytical Chemistry, Dicle University, 21280 Diyarbakir, Türkiye; 5Valorization of Natural Resources, Bioactive Molecules and Biological Analysis Unit, Department of Chemistry, University of Mentouri Constantine 1, Constantine 25000, Algeria; 6Biotechnology Research Center (C.R.Bt), Ali Mendjeli, Nouvelle Ville, UV 03 BP, Constantine P.O. Box E73, Algeria; 7Department of Biochemistry, College of Science, King Saud University, Riyadh 11451, Saudi Arabia; 8Department of Life Sciences, University of Mumbai, Vidyanagari, Mumbai 400098, India; ahmadali@mu.ac.in

**Keywords:** antidiabetic activity, *Hyoscyamus albus* L., α-glucosidase, α-amylase, molecular docking

## Abstract

This study aimed to investigate the chemical composition and antidiabetic properties of cultivated *Hyoscyamus albus* L. The ethanol extract was analyzed using LC-MS/MS, and 18 distinct phenolic compounds were identified. Among these, p-coumaric acid (6656.8 ± 3.4 µg/g), gallic acid (6516 ± 1.7 µg/g), luteolin (6251.9 ± 1.3 µg/g), apigenin (6209.9 ± 1.1 µg/g), and rutin (5213.9 ± 1.3 µg/g) were identified as the most abundant polyphenolic molecules. In the in vitro antidiabetic experiment, the ability of the plant extract to inhibit α-glucosidase and α-amylase activities was examined. The results indicated that the extract from *H. albus* L. exhibited a higher inhibitory effect on α-amylase compared to α-glucosidase, with an IC50 of 146.63 ± 1.1 µg/mL and 270.43 ± 1.1 µg/mL, respectively. Docking simulations revealed that luteolin, fisetin, and rutin exhibited the most promising inhibitory activity against both enzymes, as indicated by their high contrasting inhibition scores. To further investigate the in vivo antidiabetic effects of *H. albus* L., an experiment was conducted using STZ-induced diabetic mice. The results demonstrated that the plant extract effectively reduced the levels of cholesterol and triglycerides. These findings suggest that *H. albus* L. may have therapeutic potential for managing hyperlipidemia, a common complication associated with diabetes. This highlights its potential as a natural remedy for diabetes and related conditions.

## 1. Introduction

Diabetes mellitus (DM) is a chronic and severe disease resulting from a disturbance in glucose metabolism. It has become more prevalent in recent years and poses a serious public health dilemma worldwide [1]. Hyperglycemia, an abnormal postprandial rise in blood sugar levels, is a risk factor for the development of type 2 diabetes [2]. In addition to macrovascular disorders such as peripheral arterial disease and coronary heart disease, diabetes can cause other conditions such as peripheral neuropathy, retinopathy, and diabetic kidney disease [3]. The two primary enzymes that catalyze carbohydrate degradation are intestinal alpha-glucosidase and pancreatic alpha-amylase. Prior to being assimilated into the intestines, starch is digested by α-amylases into oligosaccharides, which are then hydrolyzed to glucose by intestinal α-glucosidase [4].

One of the treatments used to control type 2 diabetes is to limit glucose absorption by inhibiting the enzymes that catalyze the hydrolysis of polysaccharides (α-glucosidase and α-amylase) [5,6]. Acarbose is among the synthetic agents known to inhibit carbohydrate digestive enzymes and represents an appropriate therapeutic method for managing postprandial glycemia [7]. However, despite the FDA’s approval of acarbose and other glucosidase inhibitors as effective agents for reducing blood glucose levels, these molecules have been associated with severe adverse effects, including abdominal distension, diarrhea, hepatotoxicity, and flatulence [7].

In recent years, significant efforts have been made to identify new naturally efficient α-amylase and α-glucosidase inhibitors to develop new treatments. Medicinal plants are great resources for inhibitor substances [8]. Therefore, plant-based enzyme inhibitors are now highly recommended, especially in developed countries that sometimes lack access to modern therapy and due to concerns about the side effects of synthetic drugs [8].

White henbane (*Hyoscyamus albus* L.) is a Mediterranean plant member of the Solanaceae family that possesses an interesting composition containing alkaloids, terpenoids, and phenolic compounds (polyphenols, flavonoids, and tannins) [9]. This plant has been used for a long time as an analgesic and diuretic and has been traditionally associated with the treatment of insomnia, asthma, whooping cough, infectious ulcers, and epilepsy [10].

The aim of this study is to identify the phytochemical constituents of an ethanolic extract of cultivated *H. albus* using liquid chromatography with tandem mass spectrometry (LC-MS/MS). Herein, we report the in vitro enzyme inhibition tests and in vivo antidiabetic activity of the *H. albus* ethanolic extract, along with docking studies. Furthermore, the detected phenolic compounds were evaluated as ligands to assess their inhibitory efficacy against α-amylase and α-glucosidase receptors to understand their biological mechanisms.

## 2. Results

### 2.1. Total Phenolic, Flavonoid, and Tannin Contents

Table 1 presents the quantities of total phenolic compounds (TPC), flavonoid compounds (TFC), and tannin compounds (TTC) in the ethanolic extracts of cultivated *H. albus*. The TPC was determined using the gallic acid standard graph with the equation y = 0.0127x − 0.0106 (R^2^ = 0.9988). Similarly, the TFC was calculated using the quercetin calibration curve with the equation y = 0.0334x − 0.1031 (R^2^ = 0.9382). The TTC content was determined using the catechin calibration curve with the equation y = 0.133x − 0.0183 (R^2^ = 0.9663).

### 2.2. Screening and Measurement of Phenolic Compounds

Figure 1 illustrates the base peak chromatogram (BPC) of the ethanolic extract of cultivated *H. albus*, while Table 2 provides the quantitative results. A total of 18 phenolic compounds with diverse chemical structures were identified. There were significant differences in the concentrations of these chemicals. The most abundant phenolic compounds in the cultivated *H. albus* extract, as determined by LC-MS/MS, were p-coumaric acid (6656.8 ± 3.4 µg/g), gallic acid (6516 ± 1.7 µg/g), luteolin (6251.9 ± 1.3 µg/g), apigenin (6209.9 ± 1.1 µg/g), and rutin (5213.9 ± 1.3 µg/g). Additionally, the extract exhibited relatively high levels of hyperoside (2123 ± 1.2 µg/g), naringenin (923 ± 2.1 µg/g), hesperetin (883 ± 1.7 µg/g), quercetin (293 ± 6.2 µg/g), and kaempferol (212 ± 2.1 µg/g) as flavonoids.

Furthermore, the cultivated *H. albus* extract contained moderate amounts of fisetin (124.26 ± 2.15 µg/g), vanillin (76.7 ± 1.2 µg/g), and salicylic acid (43.71 ± 3.3 µg/g) and negligible amounts of tr-caffeic acid (12.71 ± 1.37 µg/g). Regarding non-phenolic substances, the extract exhibited a high concentration of quinic acid (114.7 ± 4.3 µg/g) and a lower amount of malic acid (3.6 ± 2.3 µg/g). Additionally, tannic acid (970 ± 1.6) and coumarin (0.8 ± 2.3), which are two subclasses of phenolic compounds, were present in the extract (Figure 1 and Table 2). The base peak chromatogram (BPC) of the LC-MS/MS chromatograms of a 250 ppb standard mix is presented in Appendix A. The analytical parameters of standard compounds are presented in Appendix A.

### 2.3. In Vitro Test Methods for Inhibiting Enzymes

Table 3 shows that the plant extract exhibited a higher inhibitory effect on α-amylase compared to α-glucosidase. Notably, the observed inhibitory potency of the plant extract on both enzymes is relatively similar to that of the standard molecule acarbose. Acarbose displayed IC_50_ values of 146.63 ± 1.1 µg/mL for α-amylase and 270.43 ± 1.1 µg/mL for α-glucosidase.

### 2.4. Molecular Docking Studies

Docking simulations were conducted on the 18 phenolic compounds of *H. albus* to evaluate their affinity and binding mode with two target enzymes, alpha-amylase and alpha-glucosidase. To ensure the accuracy of the docking procedure within the target enzymes, the docking of the co-crystallized inhibitor, acarbose, had beenpreviously established. Acarbose exhibited spatial conformations closely aligned with their corresponding co-crystallized structures, with root-mean-square deviation (RMSD) values below 1, validating the accuracy of the docking configuration.

The results of the docking simulations revealed that the top three phenolic compounds demonstrated favorable binding within the catalytic pocket of both tested enzymes. The theoretical binding modes of these compounds are depicted in Figure 2 and Figure 3. These specific molecules were selected based on their notable differential inhibition scores against the two enzymes, as outlined in Table 4 and Table 5.

### 2.5. Drug Similarity and the ADMET Profile

In terms of the pharmacokinetic and ADMET (absorption, distribution, metabolism, excretion, and toxicity) profile, an ideal drug candidate should comply with Lipinski’s Rule of Five, which is a set of criteria for optimal oral administration, and it should also be free of potential toxicity [11]. Therefore, conducting pharmacokinetic and ADMET profiling of the identified promising hit molecules is crucial in the drug development process to assess their bioavailability and potential adverse effects. The pharmacokinetic and drug-likeness characteristics of the top active ligands from *H. albus* are summarized in Appendix A.

### 2.6. In Vivo Antidiabetic

#### 2.6.1. Acute Oral Toxicity

Table 6 presents the results of biochemical markers, including ASAT (aspartate aminotransferase), ALAT (alanine aminotransferase), creatinine, and urea. Elevated levels of aminotransferase activity (ALAT and ASAT) are typically indicative of hepatotoxicity [12]. However, upon comparing the treated mice with the control group, no significant impact on the biochemical parameters was observed, even at the highest dosage of 2000 mg/kg body weight. These results suggest that the *H. albus* extract does not cause any hepatotoxicity. On the other hand, an increase in creatinine and urea concentrations is commonly associated with renal dysfunction [13].

The LD50 (median lethal dose) for the extract was determined to be above 2000 mg/kg, indicating a relatively safe profile. Additionally, administering various concentrations of the extract did not result in any notable changes in renal parameters and liver enzymes, suggesting that the phenolic compounds identified by LC-MS/MS are non-toxic and do not have adverse effects on internal organs at the concentrations present in the extract.

Moving on to the hypoglycemic activity, an oral glucose tolerance test (OGTT) was conducted in healthy mice, and the results are presented in Table 7. The hypoglycemic activity of the *H. albus* extract was evaluated through an oral glucose tolerance test (OGTT) conducted in healthy mice. The results, presented in Table 7, indicate a significant reduction in blood glucose levels in all experimental groups compared to the control group. The effect of the reference drug, glibenclamide, was found to be significantly greater than that of the *H. albus* extract. However, it is worth noting that the administration of 20 mg/kg of the *H. albus* extract led to a noticeable decrease in blood sugar levels in mice, as evident from the data in Table 7.

#### 2.6.2. Antidiabetic Activity in Streptozotocin-Induced Hyperglycemia Model

As shown in Table 8, *H. albus* extract significantly reduced blood glucose concentrations in the streptozotocin (STZ)-induced mice model compared to controls and glibenclamide.

After 20 days of treatment, mice administered the *H. albus* plant extract at both tested doses (10 mg/kg and 20 mg/kg) exhibited nearly identical blood glucose concentrations compared to non-diabetic mice. In contrast, the glibenclamide group (20 mg/kg) did not significantly lower blood sugar levels compared to the control group. The most significant reduction in blood glucose levels was observed at the concentration of 20 mg/kg. Therefore, after 20 days of treatment, the cultivated *H. albus* extract effectively reduced hyperglycemia in both STZ-diabetic and healthy mice, leading to the achievement of normal blood sugar levels.

In addition to hyperglycemia, diabetes is characterized by polydipsia (increased drinking), polyuria (excessive urine production), and weight loss with polyphagia (increased appetite). Table 8 displays the variations in body weight of diabetic mice from the beginning of the experiment to the final day. Compared to the normal control group, the injection of streptozotocin caused a significant decrease in body weight in all diabetic mice. However, in contrast to the normal model group, treatment with *H. albus* extracts did not significantly increase the body weight of treated mice. On the contrary, the glibenclamide-treated group experienced significant weight loss after 20 days at a concentration of 20 mg/kg.

Following streptozotocin injection, diabetic control mice exhibited a substantial increase in total serum cholesterol and triglyceride levels, along with a decrease in the high-density lipoprotein (HDL) ratio compared to normal mice (Table 8). In contrast, administration of the studied extracts at doses of 10 mg/kg and 20 mg/kg to diabetic mice resulted in a significant reduction in cholesterol and triglyceride levels, accompanied by a remarkable increase in HDL levels. These findings indicate that the extracts effectively regulated the lipid profiles, restoring them to levels comparable to those observed in normal control animals.

## 3. Discussion

The presence of phenolic compounds in medicinal plants has sparked significant interest due to their potential antioxidant effects in various diseases, including cancer, heart disorders, stroke, and inflammation [14]. LC-MS/MS has been widely utilized in quantitative applications, offering high selectivity and sensitivity compared to other LC methods [15]. In this study, LC-MS/MS analysis of cultivated *H. albus* ethanol extracts was performed, leading to the identification of 18 phenolic chemicals (Table 1).

The analysis revealed a rich profile of flavonoids in the *H. albus* ethanolic extract, including apigenin and luteolin, which have demonstrated various therapeutic properties [16]. Additionally, rutin, kaempferol, and quercetin, known for their anti-inflammatory properties, were found in relatively high concentrations in *H. albus*, suggesting its potential for mitigating and treating chronic human conditions [17]. Our findings showed a higher phenolic content compared to a previous study by Tlili [18], who employed LC-ESI-MS and identified only 11 phenolic compounds.

Our research represents a novel contribution to the field, building upon previous studies on the therapeutic effects of *H. albus*. In particular, a study by Bourebaba et al. [19] investigated the antidiabetic effects of calystegines extracted from *H. albus* seeds, highlighting their potential as antihyperglycemic and hypolipidemic agents. In our study, the *H. albus* extract demonstrated strong inhibition against α-amylase compared to α-glucosidase, which is likely attributed to the presence of phenolic compounds synthesized by the plant. Phenols have been shown to reduce blood glucose levels through various mechanisms, including the downregulation of carbohydrate digestion and intestinal glucose uptake, activation of pancreatic insulin production, stimulation of hepatic glucose release, and facilitation of glucose assimilation in insulin-sensitive tissues [20].

Regarding the α-glucosidase enzyme, the docking analysis demonstrated that the top three phenols interacted with key residues in the enzyme’s active site, namely Asp616, Asp518, Asp404, His674, and Arg600. In α-glucosidase, the basic residues His674 and Arg600 and the acidic residues Asp616, Asp518, Asp404, and Glu521 are shown to have a critical catalytic role [21]. The lead-selected compounds bind at the entrance zone of α-glucosidase’s active site (Figure 2), either acting as a competitive inhibitor by preventing the substrate from entering the active site or as an uncompetitive inhibitor by blocking the exit of the product from the enzyme [22].

With the lowest binding energy value (−8.6 kcal/mol), acarbose interacts with the α-glucosidase catalytic site by establishing 10 hydrogen bonds with the amino acids Asp404, His674, Asp616, Arg600, Met519, and Asp282. The analysis of interactions with 5NN8 predicted that the top five phenols occupy the same location as acarbose, as depicted in the expanded view of the active site in Figure 2. Their binding affinities are as follows: luteolin (−8.2 kcal/mol), fisetin (−8.2 kcal/mol), and rutin (−8.0 kcal/mol).

Based on the binding scores, the two flavonoids, luteolin and fisetin, exhibit the strongest binding to α-glucosidase. The docked view shows them deeply embedded within the binding cavity of α-glucosidase (Figure 2). According to Vaya et al. [23], the presence of hydroxyl radicals on the flavonoid B ring likely facilitates the dispersion of electronic states, enabling easier hydrogen donation and formation of hydrogen bonds with the α-glucosidase catalytic site residues.

The docking study conducted by Lim et al. [24] on selected compounds, including fisetin and other flavonoids, with α-glucosidase provides valuable insights into the binding interactions of these compounds, complementing our findings on the phenolic compounds from *H. albus*. Luteolin forms three hydrogen bonds with the residues Asp282, Arg600, and His674. Regarding fisetin, His674 and Asp404 provide hydrogen bonding interactions with this molecule (Table 3 and Figure 2).

In the case of α-amylase, substrate hydrolysis involves the participation of key residues Asp197 and Glu233, which contain carboxylic acids and form a covalent glycosyl-enzyme intermediate. Asp197 acts as the catalytic nucleophile, targeting the aglycon portion of the substrate during the formation of the covalent intermediate. Glu233, on the other hand, acts as an acid/base catalyst, facilitating the protonation of the leaving group and deprotonation of water, which breaks the covalent bond between the substrate and Asp197 and attaches an OH group to it [25]. Another important residue involved in the substrate hydrolysis mechanism is Asp300, which anchors the conjugation of bound substrates [26].

Table 5 presents the binding affinity represented by the docking score and the catalytic residue involved in hydrogen bonding for the top three phenols with α-amylase (2QV4). The co-crystallized ligand acarbose exhibited a binding energy of −9.2 kcal/mol. A detailed examination of its interaction with the amylase revealed 13 hydrogen interactions with Gly164, Thr163, Ala106, Asn105, Trp59, Val107, His101, Gln63, Arg195, Glu233, Asp300, Tyr62, and His299, which is consistent with the active residues reported earlier [27]. The top three compounds selected based on their binding score are fisetin (−9.3 kcal/mol), quercetin (−9.3 kcal/mol), and rutin (−9.1 kcal/mol). They were found to exhibit interesting modes of interaction, interfering with the three key amino acids of the enzyme’s active site through various hydrogen, hydrophobic, and van der Waals interactions, as depicted in Figure 3.

Based on the docking score and conformation, fisetin appears to have the highest affinity toward the active site of α-amylase. This ligand forms four hydrogen bonds, two of which are with the key amino acids Asp197 and Asp300. Indeed, hydrogen bond interactions with receptors are crucial as they provide the required organization for distinct folding and selectivity, facilitating intermolecular interactions within the protein–ligand complex [28]. Except for rutin, the other compounds form at least one hydrogen bond with one of the key amino acids of the catalytic triad (Asp197, Glu233, and Asp300).

The modification of α-amylase activity has significant implications for carbohydrate utilization as an energy source. Natural α-amylase inhibitors have been found in various medicinal herbs and have been explored for their potential in treating diabetes [29,30]. In previous studies, several plant compounds, including quercetin [31], luteolin [32], and fisetin [33], were identified as inhibitors of α-amylase, consistent with our findings.

For a bioactive molecule to be effectively absorbed through biological membranes, it must satisfy Lipinski’s Rule of Five criteria. These criteria include having a maximum of 5 hydrogen bond donors, 10 hydrogen bond acceptors, a molecular weight below 500 g/mol, and a LogP value (partition coefficient) not exceeding 5 [34]. Among the compounds tested, acarbose and rutin were the only ones that violated Lipinski’s rule, indicating potential limitations in their absorption and bioavailability. The other compounds in our study met Lipinski’s criteria, suggesting their favorable characteristics for biological membrane permeability and potential as bioactive molecules (Appendix A).

Indeed, aqueous solubility and log S are crucial factors that impact the oral absorption and bioavailability of a compound. These properties determine the substance’s ability to dissolve in water and interact with enzymes and transporters in the intestinal wall, which ultimately affect its absorption [35,36]. In this study, all the bioactive compounds demonstrated good solubility, falling within the appropriate range.

Caco-2 cell permeability is another important criterion used to assess the permeability of compounds across the gut–blood barrier [37]. The results indicated that the three flavonoids, luteolin, and fisetin exhibited excellent permeability, suggesting their potential for effective absorption.

Molecular flexibility, as determined by the number of rotatable bonds (nRB), is another chemical characteristic that influences the bioavailability of a molecule [37]. The selected bioactive molecules in this study had nRB values ranging from 1 to 7, which aligns with the reported range for the majority of drug molecules (1–10) [38]. Importantly, none of the promising molecules have the ability to cross the blood–brain barrier (BBB). This is a desirable characteristic for antidiabetic compounds, as it eliminates the possibility of adverse effects on the central nervous system.

Regarding their interactions with enzymes, the phenols luteolin, fisetin, and quercetin are predicted to potentially inhibit CYP1A2, CYP2C9, and CYP3A4 enzymes. Cytochrome P450 (CYP) enzymes play a vital role in metabolizing various substances in the body, including drugs and xenobiotics [39]. Inhibition of these enzymes can affect drug clearance and increase the risk of toxicity [40]. Additionally, the evaluation of toxic behavior suggests that none of the promising compounds are hERG blockers, which eliminates the risk of cardiac toxicity. However, rutin may pose a potential risk of immunotoxicity, while only quercetin, fisetin, and luteolin have the potential to behave as estrogen receptor disruptors. It is important to note that these predictions and evaluations are based on computational analysis and should be further validated through in vitro and in vivo studies for a more comprehensive understanding of the compounds’ safety and pharmacological profiles.

Before conducting pharmacological research and developing phytopharmaceutical products, it is important to determine the acute toxicity of any medicinal herb [41]. The liver and kidney are vital organs involved in metabolism and waste elimination, respectively. Biochemical assessments of these organs can provide insights into the toxic effects of substances without sacrificing animals [42]. In this study, the renal parameters and liver enzymes showed no signs of alteration in the functioning of these organs after administering varying amounts of *H. albus* extracts. These findings suggest that the plant may be a relatively safe candidate for drug development, but further toxicological research on different models and time scales is necessary to confirm its safety profile.

In terms of the oral glucose tolerance test, the results indicate that the anti-enzymatic activities of *H. albus* extracts may prevent a rise in postprandial blood glucose levels. Additionally, *H. albus* extracts are rich in phenolic compounds known for their antidiabetic effects [43]. Several phenols, such as caffeic acid, quercetin, tannic acid, and naringenin, have been shown to affect glucose uptake by inhibiting glucose carriers (sodium-glucose transporter-1) and SGLT2 (sodium-glucose transporter-2) in enterocytes [44]. These mechanisms contribute to the regulation of blood glucose levels.

The results obtained from the oral glucose tolerance test (OGTT) indicate a significant reduction in blood glucose levels in all experimental groups compared to the control group, suggesting the potential hypoglycemic activity of the *H. albus* extract. Although the effect of the reference drug, glibenclamide, was observed to be significantly greater than that of the *H. albus* extract, it is important to consider the context of this comparison. Glibenclamide is a well-established antidiabetic drug known for its potent hypoglycemic effects. Therefore, its superior performance in reducing blood glucose levels is expected. However, the observed decrease in blood sugar levels following the administration of 20 mg/kg of the *H. albus* extract is noteworthy. This suggests that the extract has the potential to exert a hypoglycemic effect, albeit to a lesser extent than the reference drug.The presence of bioactive phenolic compounds identified through LC-MS/MS analysis supports the hypothesis that these compounds may contribute to the observed hypoglycemic activity. However, it is important to acknowledge that the exact mechanisms underlying the hypoglycemic effect of *H. albus* extract require further investigation. Future studies should focus on elucidating the specific phenolic compounds responsible for the hypoglycemicactivity and exploring their mechanisms of action. Additionally, investigating the dose–response relationship and conducting long-term studies could provide valuable insights into the optimal dosage and duration of treatment.Furthermore, it is worth considering the potential synergistic effects of the different components present in the *H. albus* extract. The combination of multiple bioactive compounds might contribute to the overall hypoglycemic activity observed.

The stability of weight observed in mice administered with *H. albus* extract suggests that the synthesis of phenolic compounds by the plant may impact neuro-endocrine regulation, which is directly influenced by hyperglycemia. This implies that *H. albus* has the potential to improve weight reduction in diabetic mice by regulating blood glucose levels.

Hyperlipidemia, characterized by elevated levels of LDL, cholesterol, and triglycerides (TRG) and reduced levels of HDL, is commonly associated with diabetes. Insulin deficiency in diabetes can inhibit enzymes involved in fatty acid transformation, leading to increased lipid levels [45,46]. Insulin treatment is known to reduce plasma triglycerides and stimulate lipoprotein lipase, thereby decreasing lipid levels in diabetic individuals [47]. The tested *H. albus* extracts significantly reduced lipid levels in diabetic mice, suggesting a possible partial recovery of insulin production. Further research, including immunocytochemical staining for β-cells and insulin level testing, should be conducted to optimize and confirm these findings.

## 4. Materials and Methods

### 4.1. Instruments and Chemicals

The chemical composition of *H. albus* was investigated by LC–MS/MS technique (Shimadzu, Kyoto, Japan). All standards were obtained from Aldrich (Sigma-Aldrich, Darmstadt, Germany). The α-amylase of microbial origin was extracted from the fungi *Aspergillus oryzae*.

### 4.2. Plant Material

Seeds of *H. albus* were sourced from the Wilaya of Mila and cultivated in 3 kg pots. After one month of germination, the seedlings were transplanted individually into separate pots in a controlled greenhouse environment with irrigation maintained at half of the field capacity. Upon reaching the final growth stage in June, the plants were harvested, dried, ground into a powder, and stored for future use.

### 4.3. Total Bioactive Compound Determination (TPC, TFC, and TTC)

The total amount of phenolic compounds (TPC) was determined using the modified Folin–Ciocalteu method [48,49]. The results were expressed as micrograms of gallic acid equivalents per milligram of extract (μg GAE/mg). Gallic acid equivalents were calculated based on a standard curve generated using gallic acid as a reference compound [48]. The total flavonoid content (TFC) was determined following the method described by Topcu et al. [50]. The results were reported as micrograms of quercetin equivalents per milligram of extract (μg QE/mg). Quercetin equivalents were calculated using a standard curve constructed with quercetin as a reference compound [50]. The total tannin content (TTC) of the various extracts was calculated according to the procedure by Oueslati et al. [51]. The results were expressed in terms of milligrams of tannic acid equivalent per gram of dry plant material (mg CAE/g). Tannic acid equivalents were determined using a standard curve generated with tannic acid as a reference compound [51].

### 4.4. Plant Extract Preparation for LC-MS/MS and an Enzyme Inhibitory Test

The hydro-alcoholic extracts (70% ethanol) were prepared using 10 g of powdered plant material for each treatment. The plant material was combined with ethanol and allowed to dissolve for 24 h at room temperature. The mixture was then filtered, and the filtrate was concentrated using a rotary evaporator (Hahnvapor) at 40 °C. The resulting residue of the hydroalcoholic extract was collected and stored at 4 °C for further use.

### 4.5. Polyphenolic Detection and Quantitation

The LC–MS/MS technique with a tandem MS system in conjunction with the UHPLC (Nexera, Shimadzu Corporation, Kyoto, Japan) was used for the analysis of our samples as described by Lekmine et al. [52]. The LC–MS analysis was performed using a set of specific instruments, including LC-30AD binary pumps, a CTO-10ASvp column oven, a DGU-20A3R degasser, and a SIL-30AC autosampler. For the separation of compounds, we employed a reversed-phase C18 Inertsil ODS-4 analytical column with dimensions of 150 mm × 4.6 mm × 3 μm, operating at a temperature of 40 °C. The elution gradient involved the use of two mobile phases: mobile phase A composed of H_2_O, ammonium formate (5 mM), and formic acid (0.1%) and mobile phase B consisting of methanol, ammonium formate (5 mM), and formic acid (0.1%). The elution of compounds was achieved using a gradient of these mobile phases.

To maintain the flow rate, the solvent flow was set at 0.5 mL/min, and a fixed injection volume of 4 μL was used for sample introduction. The electrospray ionization (ESI) source was employed for air pressure ionization. To optimize the ESI conditions, we determined the following parameters: a desolvation line (DL) temperature of 250 °C, an interface temperature of 350 °C, a heat block temperature of 400 °C, a drying gas flow rate of 15 L/min, and a nebulizing gas flow rate of 3 L/min.

For quantification, a multiple-reaction monitoring (MRM) strategy was adopted, and three transitions per compound were used. The first transition served for quantification, while the subsequent transitions were utilized for confirmation purposes.

To ensure the reliability of the LC–MS/MS method, comprehensive method validation parameters were determined. The analytical characteristics of reference compounds, including linearity ranges and rectilinear regression estimates, were previously documented in the literature [53]. All calibration curves for the studied compounds exhibited linearity and reproducibility, with correlation coefficients exceeding 0.991. Additionally, the limit of detection (LOD) and limit of quantitation (LOQ) for the analyzed compounds were reported by Ertas and Yener [53]. LOD values ranged from 0.05 to 25.8 g/L, while LOQ values ranged from 0.17 to 85.9 g/L. Moreover, the recovery percentages of phenolic compounds ranged from 96.9% to 106.2%.

In this study, the identification of the analyzed molecules was carried out through a rigorous approach combining multiple techniques. Initially, library MS/MS matching was performed by comparing the acquired spectra with a comprehensive database of known phytochemicals. Additionally, relevant literature references were consulted to cross-reference retention times, mass spectra, and fragmentation patterns. Furthermore, accurate mass measurement, isotopic pattern analysis, and, whenever possible, comparison with authentic standards were employed for further confirmation. This integrated approach ensured the reliable identification of the targeted phytochemicals.

### 4.6. Enzymatic Inhibitory Assay

The enzymatic inhibitory assays were conducted in 96-well microplates, and the absorbance readings were taken using a Multimode Plate Reader. Acarbose, a well-known inhibitory molecule, was used as the standard for comparison in this study. All experiments were performed in triplicate to ensure accuracy and reproducibility of the results. For the evaluation of anti-alpha-amylase activity, the iodine/potassium iodide method [54] was employed. The absorbance was measured at 630 nm using a spectrophotometer. The inhibitory activity of the test samples was compared to that of acarbose, and an IC50 value was determined as a measure of the potency of the samples in inhibiting alpha-amylase. The alpha-glucosidase inhibitory test was carried out using the Nampoothiri technique [55]. Again, acarbose was used as a positive control in this assay. The absorbance readings were recorded, and the inhibitory activity of the test samples against alpha-glucosidase was evaluated.

### 4.7. Computational Docking Methodology

The 2D structures of 18 phenolic compounds from H. albus were retrieved in SMILES strings from the PubChem database. The Build Structure module of UCSF Chimera 1.15 was used to generate 3D structures of the compounds. To prepare the ligands for docking, hydrogen atoms were added, and atom valences were adjusted. The 3D structures of α-glucosidase (PDB ID: 5NN8) and α-amylase (PDB ID: 2QV4) complexed with the inhibitor acarbose were obtained from the Protein Data Bank (PDB) with resolutions of 2.45 Å and 1.97 Å, respectively. The protein structures were prepared using the Dock Prep wizard of UCSF Chimera software. This involved removing water molecules, other heteroatoms, and co-crystallized ligands. Charges were corrected using the AMBER ff14SB force field, and polar hydrogen atoms were added.To determine the binding sites for docking, the co-crystallized inhibitor acarbose was used as a reference. The grid generation was established based on the localization of acarbose within the active sites of both enzymes. The grid parameters, including size and spacing, were set to ensure comprehensive sampling of the binding site. Molecular docking simulations were performed using the Autodock Vina program. The scoring function in Autodock Vina estimates ligand binding affinities based on empirical potentials. The search algorithm systematically explores the conformational space of the ligands within the defined binding sites to identify favorable binding modes.

### 4.8. Pharmacokinetic and ADMET Profile

Pharmacokinetic profiling, comprising drug-likeness, was performed using the web-based application Swiss-ADME server (https://www.swissadme.ch (accessed on 5 April 2023)). The web-based applications ADMETlab 2.0 (https://admetmesh.scbdd.com/ (accessed on 5 April 2023)) and ProTox-II (https://tox-new.charite.de (accessed on 5 April 2023)) were also used to perform the toxicity evaluation. For each active compound, several parameters were calculated, including human intestinal absorption, bioavailability, brain penetration, carcinogenicity, mutagenicity, and immunotoxicity. The SMILES of the ligands were uploaded to the used computational tools, and their results were computed.

### 4.9. In Vivo Antidiabetic Activity

#### 4.9.1. Animals

Male albino mice weighing approximately 20 g and 28 g were obtained from the Paster Institute in Algeria. The mice were housed in standard climate-controlled cages with a humidity level of 50% and a temperature maintained at 25 °C. They were provided with a standard diet for a period of 10 days and then subjected to a 12h fasting period before screening.

#### 4.9.2. Acute Toxicity

The male albino mice were divided into four groups, each consisting of four mice. They were administered a single oral dose of 1000 mg/kg, 1500 mg/kg, and 2000 mg/kg of *H. albus* extract, respectively. The control group received a saline solution. Detailed observations of the mice were conducted during the first 4 h after treatment and at least twice daily for the subsequent 14 days. At the end of the study, the animals were anesthetized and sacrificed, and blood samples were collected in heparinized tubes for renal and hepatic screening tests.

#### 4.9.3. Hypoglycemic Activity

The oral glucose tolerance test was conducted on four groups of healthy mice (*n* = 4). Prior to the experiment, all animals underwent a 12h fasting period. *H. albus* extract was administered at doses of 10 mg/kg and 20 mg/kg, respectively, 30 min before oral glucose administration at a dose of 2 g/kg. In the control group, glibenclamide at a dose of 20 mg/kg was used as the reference. Blood glucose levels were measured at 0, 30, 60, 90, and 120 min following glucose administration.

#### 4.9.4. Antihyperglycemic Activity in Streptozotocin-Induced Hyperglycemia Model

Male albino mice were fasted overnight prior to the experiment and divided into five groups, each consisting of four mice. Experimental diabetes was induced by a single intraperitoneal injection of streptozotocin solution at a concentration of 130 mg/kg (dissolved in 0.1 M citrate buffer, pH 4.5). Mice that exhibited diabetes symptoms after 72 h, with blood glucose levels equal to or greater than 250 mg/dL, were selected for further examination. For a duration of 20 days, the experimental diabetic mice received the following oral treatments:Group 1: normal control mice;Group 2: diabetic control mice;Group 3: diabetic mice treated with *H. albus* at a concentration of 10 mg;Group 4: diabetic mice treated with *H. albus* at a concentration of 20 mg;Group 5: diabetic mice administered glibenclamide at a dose of 20 mg/kg.

Blood glucose levels and body weight changes were measured every five days. At the end of the experiment, all animals were anesthetized and sacrificed, and blood samples were collected to determine the levels of total cholesterol, HDL, and triglycerides.

## 5. Conclusions

This research investigates the antidiabetic activity of *Hyoscyamus albus*, a medicinal herb of pharmaceutical involvement. To our knowledge, this is the first research that reported the efficacy of *H. albus* ethanol extract as an inhibitor of the two regulatory enzymes α-amylase and α-glucosidase, which are implicated in diabetes. The combination of the in vitro and in vivo tests confirms the great potential of *H. albus* for inhibiting these key enzymes and demonstrates its antihyperglycemic action. The insilico results agreed with the in vitro and in vivo findings and produced clear evidence of the strong molecular interactions between the phytochemicals and α-amylase and α-glucosidase. Despite this, a possibility of a synergistic inhibitory effect between the different compounds should not be neglected because this can have a more significant inhibition impact with a different mode of interaction.

In the present study, flavonoids, particularly fisetin and rutin, exhibited significant binding affinity with both enzymes. The insights derived from this research have the potential to contribute to the development of new functional foods rich in flavonoids for the treatment of diabetes. These functional foods could target the inhibition of both α-amylase and α-glucosidase, aiming to address metabolic dysregulation associated with diabetes.

Overall, our study focused on investigating the phenolic compounds present in *H. albus* using a negative-mode LC-MS/MS approach. While this analysis provided valuable insights into the phenolic profile, we acknowledge the limitations of our approach in not exploring the alkaloid composition. Further investigations employing alternative techniques are necessary to comprehensively elucidate the complete phytochemical profile of *H. albus*, including its alkaloid constituents. Future studies addressing these aspects will contribute to a more comprehensive understanding of the phytochemical composition and its potential therapeutic applications. Furthermore, our study made a significant contribution to the understanding of *H. albus* and its potential hypoglycemic activity. However, it is important to recognize the limitations associated with the small sample size employed in our study. The sample size of *n* =4 in each group may have limited the statistical power and generalizability of our findings. To address this limitation, future research endeavors should aim to conduct studies with larger sample sizes, enabling a more robust statistical analysis and enhancing the reliability and applicability of the results.

## Figures and Tables

**Figure 1 pharmaceuticals-16-01015-f001:**
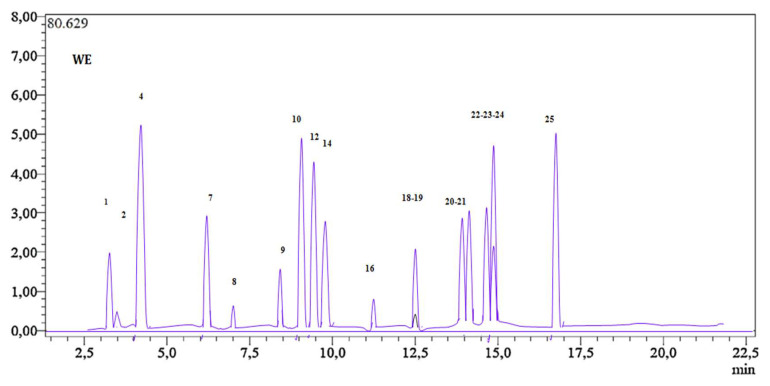
LC-MS/MS chromatogram of cultivated *H. albus* ethanolic extract.

**Figure 2 pharmaceuticals-16-01015-f002:**
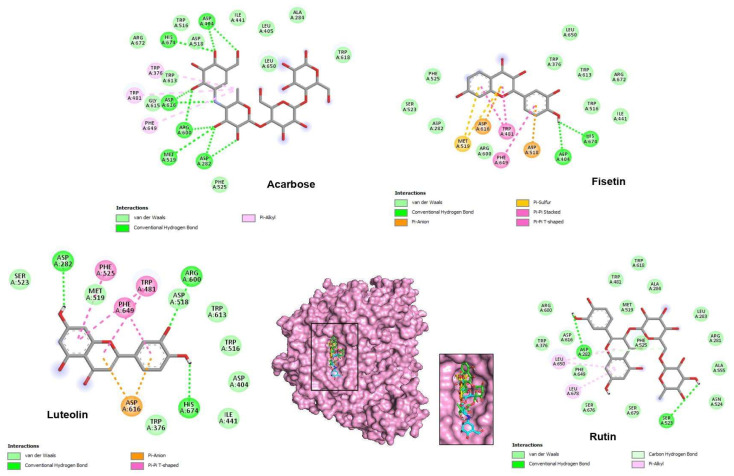
Two-dimensional predicted binding mode for acarbose and the best top phenolic ligands of *H. albus* with α-glucosidase (5NN8). In the center, a 3D view of the binding gorge with an expanded picture at the active site showing acarbose, fisetin, luteolin, and rutin as cyan, yellow, orange, and green sticks, respectively, bound to the surface of α-glucosidase.

**Figure 3 pharmaceuticals-16-01015-f003:**
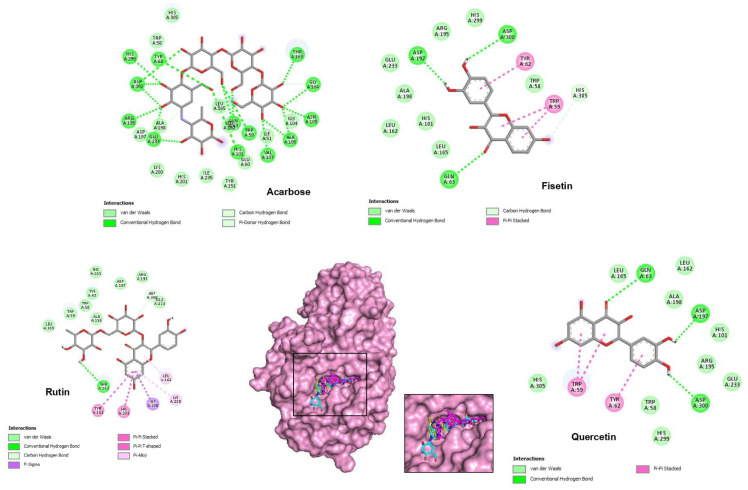
Two-dimensional predicted binding mode for acarbose and the five best top polyphenolic ligands of *H. albus* with α-amylase (2QV4). In the center, a 3D view of the binding gorge with an expanded picture at the active site shows acarbose, fisetin, rutin, and quercetin as cyan, bleu, magenta, and yellow sticks, respectively, bound to the surface of α-amylase.

**Table 1 pharmaceuticals-16-01015-t001:** Contents of the ethanolic extract of *H. albus* in terms of total phenols, flavonoids, and tannins.

Sample	TPC (μgEAG/mg E)	TFC (μg EQ/mgE)	TTC (μg ECT/mgE)
*H. albus* ethanolic extract	245.20 ± 0.53 a	120.55 ± 0.56 a	60 ± 0.42 b

Significant differences between the samples are denoted by letters (a, b) (*p* < 0.05), indicating statistically significant variations among the groups.

**Table 2 pharmaceuticals-16-01015-t002:** Amounts (μg/g dry plant) of selected phytochemicals in cultivated *H. albus* ethanolic extract.

Compound Number		RT	[M-H]^−^	MS2 (Collision Energy)	Cultivated *H. albus* (µg dry/g Extract)
1	Quinic acid	3.32	191.0	85 (22), 93 (22)	114.7 ± 4.3
2	Malic acid	3.54	133.1	115 (14), 71 (17)	3.6 ± 2.3
3	tr-Aconiticacid	4.13	172.9	85 (12), 129 (9)	N.D.
4	Gallic acid	4.29	169.1	125 (14), 79 (25)	6516 ± 1.7
5	Chlorogenic acid	5.43	353.0	191 (17)	N.D.
6	Protocatechuicacid	5.63	153.0	109 (16), 108 (26)	N.D.
7	Tannic acid	6.46	183.0	124 (22), 78 (34)	970.0 ±1.6
8	tr-Caffeic acid	7.37	179.0	135 (15), 134 (24), 89 (31)	12.71 ± 1.3
9	Vanillin	8.77	151.1	136 (17), 92 (21)	76.7 ± 1.2
10	*p*-Coumaric acid	9.53	163.0	119 (15), 93 (31)	6656.8 ± 3.4
11	Rosmarinic acid	9.57	358.9	161 (17), 133 (42)	N.D.
12	Rutin	10.16	608.1	300 (37), 271 (51), 301 (38)	5213.9 ± 1.3
13	Hesperidin	9.69	611.1	303, 465	N.D.
14	Hyperoside	10.53	459.1	300, 301	2123.0 ± 1.2
15	4-OHBenzoicacid	11.72	137.0	93, 65	N.D.
16	Salicylicacid	11.72	137.0	93, 65, 75	43.71 ± 3.3
17	Myricetin	11.94	317.0	179, 151, 137	N.D.
18	Fisetin	12.61	285.0	135, 121	124.26 ± 2.1
19	Coumarin	12.52	147.0	103, 91, 77	0.8 ± 2.3
20	Quercetin	14.48	300.9	179, 151, 121	293.0 ± 6.2
21	Naringenin	14.66	271.0	151, 119, 107	923.0 ± 2.1
22	Hesperetin	15.29	301.0	164, 136, 108	883.0 ± 1.7
23	Luteolin	15.43	285.0	175, 151, 133	6251.9 ± 1.3
24	Kaempferol	15.43	285.0	217, 133, 151	212.0 ± 2.1
25	Apigenin	16.31	269.0	151, 117	6209.9 ± 1.1

N.D.: Compound not detected above the LOD.

**Table 3 pharmaceuticals-16-01015-t003:** In vitro antidiabetic assay of *H. albus* extracts.

Sample	IC_50_ of Enzyme Inhibitory Assay (µg/mL)	
	Alpha-Amylase	Alpha-Glucosidase
Cultivated *H. albus*	120.5 ± 1.3	243.2 ± 1.3
Acarbose	146.63 ± 1.1	270.43 ± 1.1

The values are calculated by the mean and standard deviation of five independent experiments.

**Table 4 pharmaceuticals-16-01015-t004:** The topthree results for docking *H. albus* phenolic ligands with α-glucosidase.

	Binding Force (Energy) (kcal/mol)	Contacts with Hydrogen (Interactions)	Hydrophobic Interactions	Van der Waals Interactions	Electrostatic Interactions
Acarbose	−8.6	Asp404, His674, Asp616, Arg600, Met519, Asp282	Trp376, Trp481, Phe649	Trp618, Ala284, Leu405, Leu650, Ile441, Trp516, Asp518, Arg672, Trp613, Gly615, Phe525	-
Luteolin	−8.2	Asp282, Arg600, His674	Phe525, Phe649, Trp481	Ser523, Met519, Trp376, Ile441, Asp404, Trp516, Trp613, Asp518	Asp616
Fisetin	−8.2	His674, Asp404	Trp481, Phe649	Ile441, Trp516, Arg672, Trp613, Trp376, Leu650, Phe525, Ser523, Asp282, Arg600	Asp616, Met519, Asp518
Rutin	−8.0	Asp282, Ser523	Leu678, Leu650	Asn524, Ala555, Arg281, Leu283, Ala284, Phe525, Met519, Trp481, Trp618, Arg600, Asp616, trp376, Phe649, Ser676	-

**Table 5 pharmaceuticals-16-01015-t005:** The topthree results for the docking of *H. albus* phenolic ligands with α-amylase.

	Binding Force (Energy) (kcal/mol)	Contacts with Hydrogen (Interactions)	Hydrophobic Interactions	Van der Waals Interactions
Acarbose	−9.2	Gly164, Thr163, Ala106, Asn105, Trp59, Val107, His101,Gln63, Arg195, Glu233, Asp300, Tyr62, His299	-	His305, Trp58, Ala198, Asp197, Lys200, His201, Ile235, Tyr151, Leu162, Leu165, Ile51, Gly104
Fisetin	−9.3	Asp197, Asp300, Gln63, His305	Tyr62, Trp59	Trp58, His299, Arg195, Glu233, Ala198, Leu162, His101, Leu165
Quercetin	−9.2	Gln63, Asp197, Asp300	Tyr62, Trp59	Leu162, Leu165, Ala198, His101, Arg195, Glu233, Trp58, His299, His305
Rutin	−9.1	Thr163	Tyr151, His201, Ile235, Lys200, Leu162	Leu165, Trp59, Trp58, Ala198, Tyr62, His101, Asp197, Arg195, Asp300, Glu233

**Table 6 pharmaceuticals-16-01015-t006:** Blood examination of mice, 14 days following treatment with *H. albus* extracts.

Groups		Parameters for the Renal Function		
	ASAT	ALAT	Urea (g/L)	Creatinine (mg/L)
Group 1 (1000 mg/kg)	156 ± 2.2	47.2 ± 3.5	0.34 ± 1.3	<0.37
Group 2(1500 mg/kg)	159.3 ± 1.2	49.6 ± 3.6	0.43 ± 1.3	<0.37
Group 3 (2000 mg/kg)	162.5 ± 1.4	48.3 ± 2.5	0.42 ± 1.5	<0.37
Control	157.2 ± 5.2	50.0 ± 2.3	0.59 ± 1.4	<0.37

ASAT:Aspartate-Amino-Transferase; ALAT: Alanine-Amino-Transferase.

**Table 7 pharmaceuticals-16-01015-t007:** Effect of *H. albus* extract on the oral glucose tolerance test (OGTT) in good-health mice.

	Amount of Blood Glucose (g/L)				
Groups	0 min	30 min	1 h	1 h and 30 min	2 h
*H. albus* 10 mg/kg	0.93 ± 0.7	2.4 ± 1.3	2.1 ± 1.7	1.9 ± 2.0	1.8 ± 1.3
*H. albus* 20 mg/kg	0.93 ± 1.1	2.01 ± 1.8	1.5 ± 2.2	0.9 ± 1.2	0.8 ± 0.6
Glibenclamide20 mg/kg	0.95 ± 0.8	0.62 ± 0.5	0.56 ± 0.2	0.47 ± 0.5	0.41 ± 0.3
Control	0.9 ± 1.2	2.3 ± 1.5	2.01 ± 2.1	1.82 ± 2.1	1.17 ± 1.4

**Table 8 pharmaceuticals-16-01015-t008:** Effect of oral administration of *H. albus* extract on hyperglycemia in STZ-diabetic mice.

Groups	Amount of Blood Glucose (g/L), Body Weight (g), and Lipid Profile												
	0 Day		5 Day		10 Day		15 Day		20 Day				
	BGL	BW	BGL	BW	BGL	BW	BGL	BW	BGL	BW	TC	HDL	TG
*H. albus* 10 mg/kg	3.97	26.3	2.61	25.2	2.52	25.6	2.32	25.3	2.11	25.3	0.85 ± 1.2	0.45 ± 1.1	0.46 ± 1.3
*H. albus* 20 mg/kg	2.86	27.4	1.59	27.6	1.65	26.5	1.24	27.9	1.32	26.6	0.91 ± 2.2	0.51 ± 2.1	0.41 ± 2.3
Glibenclamide 20 mg/kg	3.61	27.5	3.93	26.3	3.43	26.1	2.91	25.9	2.82	25.3	/	/	/
Untreated diabetic mice	3.51	28.2	4.53	24.3	4.31	22.1	4.62	22.6	4.35	22.3	1.83 ± 1.6	0.21 ± 1.4	0.89 ± 1.1
Normal	1.08	26.6	1.02	26.5	0.98	26.1	1.12	26.8	1.10	27.4	0.95 ± 1.3	0.51 ± 2.2	0.34 ± 1.6

BGL: Blood glucose level; BW: Body weight; HDL: High-density lipoprotein; TC: Total cholesterol; TG: Triglycerides.

## Data Availability

Data is available within article and Appendix A.

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
