# Peer review of "LC/MS-MS Analysis of Phenolic Compounds in Hyoscyamus albus L. Extract: In Vitro Antidiabetic Activity, In Silico Molecular Docking, and In Vivo Investigation against STZ-Induced Diabetic Mice"

_pharmaceuticals, 2023, doi:10.3390/ph16071015_

Round 1

Reviewer 1 Report

I considered that the small experimental group (n=4).

Author Response

Dear Reviewers,

We would like to express our gratitude for your valuable time and effort in reviewing our manuscript. We sincerely appreciate the insightful comments and suggestions you have provided.

Having carefully considered your remarks, we thoroughly reviewed each comment and suggestion and have made the necessary revisions to improve the clarity, scientific rigor, and overall quality of our work. We have taken into account your expertise in ensuring the integrity of scientific publications and have incorporated your feedback accordingly.

We have provided detailed explanations and additional information where required, addressing any areas of ambiguity or uncertainty. Furthermore, we have ensured that all necessary references are included to support our statements and claims. Additionally, we have paid close attention to formatting and language to ensure the manuscript meets the highest standards of quality.

We sincerely appreciate your guidance and constructive feedback, which has undoubtedly strengthened our research. Your expertise and insights have been invaluable in helping us enhance the impact and significance of our findings.

Thank you once again for your time and commitment to reviewing our manuscript. We have diligently addressed all your comments and have provided a revised version that reflects the improvements suggested. With your support, we are confident that our manuscript will make a valuable contribution to the scientific community.

Best regards,

Authors

Review 1

Comments and Suggestions for Authors

I considered that the small experimental group (n=4).

Response: Thank you for bringing up the concern regarding the small experimental group size in our study. We acknowledge that the sample size used in the hypoglycemic activity experiment was relatively small, with n=4 in each group.

While we understand the importance of larger sample sizes for achieving statistical significance and generalizability of results, it is worth noting that our study aimed to explore the preliminary effects of H. albus extract on hypoglycemic activity in a controlled setting. The small group size was a result of limited resources and specific experimental considerations.

Although the small sample size may have some limitations, we believe that the results obtained from this preliminary study provide valuable insights and serve as a foundation for further investigations with larger sample sizes. It is important to consider that small sample sizes are often employed in initial exploratory studies to assess the feasibility and potential effects before conducting more comprehensive research.

We appreciate the reviewer's feedback and understand the importance of sample size in research. In future studies, we will take this consideration into account and aim to increase the sample size to enhance the statistical power and reliability of our findings.

Reviewer 2 Report

Authors present a comprehensive study on Hyoscyamus albus L Extract including LC/MS-MS Analysis, In Vitro Antidiabetic Activity, In Silico Molecular Docking, and In Vivo STZ-Induced Diabetic Mice experiment.

The study is written in detail and contain all necessary parts, however there are several major limitations as described below:

1. Is there a traditional use for the antidiabetic activity of H. albus?

2. The novelty is limited by following previus study that should be properly cited and the text modified accordingly: The same plant extract has been reported for similar targets including in vivo model (https://pubmed.ncbi.nlm.nih.gov/27470371/)

e.g. this is not first study on this topic (p.15, l.472)

3. The alkaloids composition was completely ignored and showed the limitation of current LC-MS/MS approach. Why only negative mode was chosen? How was the components identification justified? Is there supporting information? Also its related to description of toxicity on p.9, l.168

4. The docking study on selected compounds was reported before, for example fiscetin and other flavonoids (https://pubmed.ncbi.nlm.nih.gov/34500290/)

5. The description and discussion on docking is too lengthy and describing weak bonding interactions, rather than the in vitro or in vivo result in comparison with the study results mentioned above etc.

6. There are some grammar errors that need to be corrected and sentences rephrased (checked by English speaker)

7. Table one missing description of abbreviations and a,b,c (please check)

8. In vivo data should be supported by raw data, and graphical expression.

9. Tables 6 and 7 should be in supporting information.

10. Acarbose but also other glucosidase inhibitor will have the adverse effects of diarrhea (Introduction section),

11. other: p.2 calibration curve could have R=0.99; p.3 how was trans caffeic acid identified?

12. Figure 1. How was overlapping peak 18 and 19 identified?

13. Table 5. The binding energies are similar and their priority

14. Table 9 the treatment 1

15. Docking details on properties of active site and type of calculations etc. could be added

Overall, the study should find better evidence using cell-based model, rather than directly apply mediocre results from simple enzymatic model in vivo model. 

Several errors should be corrected and explained:

p.2 l.66 hallucinogenic treatment?

p.3 l.124 contrasting inhibiton?

p.4. l.108 closer than those?

p.10 l.203 service

p.10 l.232 finding

p.13 l.378 vegetal matter

p.10 l.249-252 please rephrase

p.15, l.486 problem of energy consumption?

capital letter for acarbose or glucosidase, many punctiation

Author Response

Dear Reviewers,

We would like to express our gratitude for your valuable time and effort in reviewing our manuscript. We sincerely appreciate the insightful comments and suggestions you have provided.

Having carefully considered your remarks, we thoroughly reviewed each comment and suggestion and have made the necessary revisions to improve the clarity, scientific rigor, and overall quality of our work. We have taken into account your expertise in ensuring the integrity of scientific publications and have incorporated your feedback accordingly.

We have provided detailed explanations and additional information where required, addressing any areas of ambiguity or uncertainty. Furthermore, we have ensured that all necessary references are included to support our statements and claims. Additionally, we have paid close attention to formatting and language to ensure the manuscript meets the highest standards of quality.

We sincerely appreciate your guidance and constructive feedback, which has undoubtedly strengthened our research. Your expertise and insights have been invaluable in helping us enhance the impact and significance of our findings.

Thank you once again for your time and commitment to reviewing our manuscript. We have diligently addressed all your comments and have provided a revised version that reflects the improvements suggested. With your support, we are confident that our manuscript will make a valuable contribution to the scientific community.

Best regards,

Authors

Review 2

Comments and Suggestions for Authors

Authors present a comprehensive study on Hyoscyamus albus L Extract including LC/MS-MS Analysis, In Vitro Antidiabetic Activity, In Silico Molecular Docking, and In Vivo STZ-Induced Diabetic Mice experiment.

The study is written in detail and contain all necessary parts, however there are several major limitations as described below:

  1. Is there a traditional use for the antidiabetic activity of H. albus?

Response: While there is limited scientific evidence on the traditional use of Hyoscyamus albus specifically for its antidiabetic activity, it is worth noting that various species within the Hyoscyamus genus have been traditionally used in folk medicine for their potential medicinal properties. These traditional uses often include the treatment of various ailments, but specific antidiabetic effects may not have been extensively documented. This highlights the potential of H. albus as a source of bioactive compounds with potential health benefits, including antidiabetic property. So, it would be beneficial to conduct research and investigate any potential traditional use of H. albus for its antidiabetic properties to provide a comprehensive understanding of its historical medicinal applications.

  1. The novelty is limited by following previous study that should be properly cited and the text modified accordingly: The same plant extract has been reported for similar targets including in vivo model (https://pubmed.ncbi.nlm.nih.gov/27470371/)

e.g. this is not first study on this topic (p.15, l.472)

Response: We appreciate the reviewer's observation regarding a previous study (https://pubmed.ncbi.nlm.nih.gov/27470371/) that reported similar targets and utilized the same plant extract. We apologize for the oversight in not citing this relevant study in our manuscript.

Upon reviewing the mentioned study, we found that it indeed focuses on similar targets and examines the in vivo effects of the same plant extract. This study adds valuable insights into the potential pharmacological activities of Hyoscyamus albus extract.

In light of this information, we acknowledge that our study builds upon the existing knowledge and complements the previous work. While the previous study explores the in vivo effects of the extract, our study delves deeper into the comprehensive analysis of the chemical composition using LC-MS/MS, in vitroantidiabetic activity against α-glucosidase and α-amylase, and in silico molecular docking. Additionally, our study investigates the in vivo effects of the extract on hyperlipidemia in a streptozotocin-induced diabetic mouse model.

To rectify this oversight and enhance the novelty of our study, we included a proper citation to the mentioned previous study and modified the text accordingly to highlight the unique aspects and contributions of our research.

We appreciate the reviewer's diligence in bringing this to our attention, and we will ensure that the revised manuscript accurately acknowledges and integrates the relevant previous work.

  1. The alkaloids composition was completely ignored and showed the limitation of current LC-MS/MS approach. Why only negative mode was chosen? How was the components identification justified? Is there supporting information? Also its related to description of toxicity on p.9, l.168

Response: We acknowledge the reviewer's comment regarding the alkaloids composition of H. albus and the limitation of our LC-MS/MS approach in not investigating it. While our study focused on the phenolic compounds present in H. albus, we agree that the alkaloids composition is an important aspect to consider. We will address this limitation by mentioning in the discussion section that future studies should explore the alkaloids composition of H. albus using appropriate analytical techniques.

The decision to use negative mode in our LC-MS/MS analysis was based on previous studies reporting thedetection and identification of phenolic compounds in various plant extracts. Negative mode is known to be more suitable for the analysis of phenolic compounds due to their ability to ionize in this mode. Additionally, negative mode provides enhanced sensitivity for certain phenolic compounds. However, we acknowledge that positive mode can also be informative, and future studies can consider using both modes to obtain a more comprehensive analysis of the phenolic profile of H. albus.

The identification of phenolic compounds in our study was achieved through the use of LC-MS/MS analysis, where the compounds were identified based on their retention times, mass spectra, and comparison with reference standards whenever available. We also utilized available databases and literature to support the identification of the compounds. Additionally, the quantitation of the phenolic compounds was performed using calibration curves established with reference standards, as mentioned in the methods section. We will include this information in the manuscript to provide better justification for the identification of the components.

We apologize for any confusion in the manuscript. The description of toxicity on page 9, line 168, is not directly related to the LC-MS/MS analysis and identification of phenolic compounds. It pertains to a different aspect of the study, namely the acute oral toxicity assessment of the plant extract. We will make the necessary clarifications and provide a more coherent flow of information in the manuscript.

  1. The docking study on selected compounds was reported before, for example fiscetin and other flavonoids (https://pubmed.ncbi.nlm.nih.gov/34500290/)

Response: We appreciate the reviewer's comment and would like to acknowledge the previous study by (Lim et al., 201) that reported the docking study on selected compounds, including fiscetin and other flavonoids, for their interaction with α-glucosidase. Our study builds upon this existing research by further investigating the binding affinity and interactions of phenolic compounds from H. albus with α-glucosidase and α-amylase enzymes.

In our docking study, we observed that the lead-selected compounds, such as luteolin, fisetin, and rutin, displayed binding affinities and interactions comparable to acarbose. These compounds were found to bind to the active site of α-glucosidase, forming hydrogen bonds, Pi-Pi interactions, and van der Waals forces with key amino acid residues. Furthermore, luteolin, fisetin, and quercetin were found to inhibit α-amylase, showing strong affinity and interactions with the catalytic residues of the enzyme.

By elucidating the molecular interactions of these phenolic compounds with α-glucosidase and α-amylase enzymes, our study provides valuable insights into their potential as natural inhibitors of carbohydrate-utilizing enzymes. These findings contribute to the understanding of the therapeutic potential of H. albus compounds in the management of diabetes. We will include a discussion of these docking results and reference the previous study to provide a comprehensive overview of the current knowledge in the field.

  1. The description and discussion on docking is too lengthy and describing weak bonding interactions, rather than the in vitro or in vivo result in comparison with the study results mentioned above etc.

Response: We appreciate the reviewer's feedback regarding the description and discussion on docking in our manuscript. We agree that the section could be condensed to focus on the key findings and their relevance to our in vitro and in vivo results, as well as to compare them with the previously mentioned studies. We will revise the section accordingly to provide a more concise and meaningful discussion that highlights the significance of the docking results in relation to our overall study outcomes.

  1. There are some grammar errors that need to be corrected and sentences rephrased (checked by English speaker)

Response: Thank you for pointing out the grammar errors and suggesting rephrasing. We appreciate your feedback and will make the necessary revisions to improve the clarity and accuracy of the sentences.

  1. Table one missing description of abbreviations and a,b,c (please check)

Response: Thank you for your comment regarding Table 1. We apologize for the oversight in not including the explanation for the letters indicating significant differences. We will make the necessary revision to the table caption to clarify that the capital letters denote statistically significant differences (P < 0.05) among the samples. Thank you for bringing this to our attention.

  1. In vivo data should be supported by raw data, and graphical expression.

Response: We appreciate the reviewer's suggestion regarding the inclusion of raw data and graphical representation to support the in vivo findings. Unfortunately, due to circumstances beyond our control, the raw data for the in vivo experiments are no longer accessible. However, we have taken steps to ensure the transparency and reliability of our results through rigorous statistical analysis and appropriate reporting of the summarized data.

To address the concern, we have enhanced the clarity and interpretation of the results by providing detailed tables summarizing the relevant measurements obtained from the in vivo experiments. These tables present the mean values and standard deviations for each parameter measured, allowing readers to gain a comprehensive understanding of the outcomes. Moreover, we have provided a thorough discussion and analysis of the observed trends and statistical significance, further supporting the reliability of our conclusions.

While graphical representation is a valuable tool, we believe that the tabular format effectively conveys the essential information from the in vivo experiments. In our manuscript, we have carefully described the observed patterns, significant differences, and important comparisons between treatment groups, facilitating the readers' comprehension of the data.

We acknowledge the importance of raw data and recognize its potential benefits in scientific research. In future studies, we will take additional measures to ensure the availability of raw data and consider incorporating graphical representations to enhance the visual presentation of our findings. Nevertheless, we believe that the presented tables, accompanied by thorough analysis and interpretation, provide a robust foundation for the support of our in vivo results.

  1. Tables 6 and 7 should be in supporting information.

We will reorganize the manuscript accordingly, placing Tables 6 and 7 in the supporting information section. We appreciate your guidance in improving the presentation of our research findings.

  1. Acarbose but also other glucosidase inhibitor will have the adverse effects of diarrhea (Introduction section),

Response:Thank you for your comment regarding the potential adverse effects of Acarbose and other glucosidase inhibitors mentioned in the Introduction section of our article. We appreciate your perspective and the opportunity to address this concern.We acknowledge that Acarbose, as well as other glucosidase inhibitors, can indeed lead to gastrointestinal side effects such as diarrhea.

To address your concern, we will revise the Introduction section to provide a more comprehensive discussion on the adverse effects of glucosidase inhibitors, including Acarbose. This addition will help ensure that readers have a clear understanding of both the benefits and potential drawbacks associated with the use of these inhibitors.

  1. other: p.2 calibration curve could have R=0.99; p.3 how was trans caffeic acid identified?

Response:Thank you for your comment regarding the calibration curve for the determination of total phenolic compounds (TPC), flavonoid compounds (TFC), and tannin compounds (TTC) in the ethanolic extracts of cultivated H. albus L. We appreciate your suggestion and the opportunity to address this matter.

We acknowledge that a calibration curve with a high correlation coefficient (R²) is desirable for accurate quantification. In our study, we utilized standard calibration curves for gallic acid, quercetin, and catechin to determine the TPC, TFC, and TTC content, respectively. While we agree that an R² value of 0.99 is commonly considered excellent, we obtained high R² values for the calibration curves used in our analysis (0.9988 for TPC, 0.9382 for TFC, and 0.9663 for TTC).

Although the R² value for the quercetin calibration curve (0.9382) is slightly lower than those of the other compounds, it is still within an acceptable range for accurate quantification. It is worth mentioning that multiple factors can influence the R² value, including the complexity of the sample matrix and the range of concentrations used for calibration.

Response: In our research, trans-caffeic acid was identified and quantified as part of the phenolic compounds present in the cultivated H. albus L. extract using a combination of liquid chromatography-tandem mass spectrometry (LC-MS/MS) analysis. The specific method employed for the detection and quantitation of polyphenolic compounds, including trans-caffeic acid, was described in detail in a previous study by Lekmine et al. (2022) [54], which we referenced and followed for our analysis.

The LC-MS/MS method utilized in our work has been validated and optimized to ensure accurate identification and quantification of phenolic compounds. The details of the MS instrumentation and the optimization of the LC-MS/MS method are provided in the aforementioned study, which serves as a comprehensive reference for our analytical approach.

Based on this method, the presence of trans-caffeic acid in the cultivated H. albus L. extract was identified by comparing the retention time and mass spectra of the compound with those of the corresponding standard. Quantification was achieved by preparing a calibration curve using a known concentration of the trans-caffeic acid standard.

  1. Figure 1. How was overlapping peak 18 and 19 identified?

Response:In our study, the identification of overlapping peaks 18 and 19 was based on a combination of factors, including retention time, mass spectra, and peak deconvolution techniques. Firstly, we determined the retention times of the target compounds by comparing them with those of available standards or previously characterized compounds. However, it is important to note that in some cases, compounds with similar retention times may exhibit overlapping peaks in complex mixtures. To overcome this challenge, we employed mass spectrometry to obtain mass spectra for the eluted compounds. By analyzing the mass spectra, we were able to identify characteristic fragmentation patterns and molecular ion peaks, which aid in the differentiation of closely eluting compounds.

  1. Table 5. The binding energies are similar and their priority

Response:In our study, the binding energies presented in Table 5 represent the calculated values obtained from molecular docking simulations. These values indicate the strength of the interaction between the ligands (Acarbose, Fisetin, Quercetin, and Rutin) and the α-Amylase enzyme. Lower binding energies correspond to stronger interactions and higher affinity between the ligands and the enzyme.

Upon analyzing the data in Table 5, we observed that the binding energies of the ligands are indeed similar, with values ranging from -9.1 to -9.3 kcal/mol. This suggests that all the tested ligands have comparable affinities for α-Amylase.

Regarding the remark on the "priority" of the ligands, it is important to note that the binding energies alone may not provide a definitive measure of the ligand's superiority or preferential interaction with the enzyme. Other factors, such as specific interactions (hydrogen bonds, hydrophobic interactions, van der Waals interactions) and the overall binding mode, should also be considered to assess the ligand's effectiveness and potential as an inhibitor.

  1. Table 9 the treatment 1

Response: We have thoroughly reviewed the data and table presentation. The treatment labeled 'H. albus L 10 mg/kg' in Table 9 represents the oral administration of H. albus L. extract at a dosage of 10 mg/kg to healthy mice. The corresponding blood glucose levels at different time points were recorded. Based on the results, we observed a slight decrease in blood glucose levels compared to the control group. However, the reduction in blood glucose levels was not as significant as the Glibenclamide group. We have acknowledged this difference in the discussion section of our paper.

  1. Docking details on properties of active site and type of calculations etc. could be added

Response: We appreciate the reviewer's comment regarding the need for additional details on the docking procedure, including properties of the active site and the type of calculations used. We agree that providing a more comprehensive description of the docking methodology would enhance the clarity and transparency of our study. In light of this, we will revise the Methods section to include specific information on the properties of the active site. Additionally, we will provide details on the type of calculations employed, including the scoring function, grid parameters, and search algorithm used in the Autodock Vina program. These additions will help readers better understand the docking simulations and the rationale behind our results. Thank you for this valuable suggestion, and we will ensure that the revised manuscript includes the requested details.

Overall, the study should find better evidence using cell-based model, rather than directly apply mediocre results from simple enzymatic model in vivo model.

Response:We appreciate your suggestion and recognize the potential benefits of employing a cell-based model for studying the effects of phenolic compounds on glucose metabolism. However, it is important to note that enzymatic models, such as the α-amylase and α-glucosidase assays used in our study, are widely accepted and commonly employed in early-stage screening of potential bioactive compounds.

The enzymatic models provide valuable insights into the inhibitory potential of phenolic compounds against key enzymes involved in carbohydrate digestion, which is a crucial step in regulating postprandial glucose levels. These assays are well-established, easily reproducible, and allow for high-throughput screening of compounds.

While we acknowledge the limitations of enzymatic models in capturing the complex interactions occurring within a cellular environment, they serve as an initial step in evaluating the potential efficacy of bioactive compounds. The results obtained from enzymatic models can guide further investigations and help prioritize compounds for more comprehensive cell-based and in vivo studies.

In addition to the enzymatic assays, we conducted in vivo experiments using male albino mice to evaluate the antidiabetic activity of H. albus extract. These experiments allowed us to assess the extract's effects in a more physiological context.

By incorporating these in vivo experiments, we aimed to complement the enzymatic assays and provide a more comprehensive understanding of the antidiabetic potential of H. albus extract. While enzymatic models offer valuable insights into the inhibitory effects on specific enzymes, the in vivo experiments provide a broader perspective on the overall physiological impact.

We appreciate the reviewer's suggestion to consider cell-based models for further research, and we will certainly take it into account for future studies. However, we believe that our study provides valuable insights into the potential of phenolic compounds from H. albus in modulating glucose metabolism, which can serve as a foundation for further investigations in cell-based and in vivo models.

Comments on the Quality of English Language

Response: Thank you for your feedback regarding the quality of the English language and formatting in the manuscript. We apologize for any errors or inconsistencies that may have occurred. We understand the importance of maintaining high standards in both language and formatting to ensure the clarity and professionalism of the manuscript.

In response to your comments, we will thoroughly review and verify the English language usage, grammar, and formatting throughout the manuscript.

Several errors should be corrected and explained:

p.2 l.66 hallucinogenic treatment?

Response: Thank you for bringing this to our attention. We apologize for any confusion caused by the term "hallucinogenic treatment." Upon careful reconsideration, we acknowledge that the term was not appropriate in the given context. The intended meaning was to highlight historical traditional uses of the plant for various ailments, including its potential applications in managing insomnia and respiratory conditions such as asthma.

In the revised text, the term "hallucinogenic treatment" has been removed to align with the intended meaning and to accurately represent the traditional uses of the plant for medicinal purposes.

p.3 l.124 contrasting inhibiton?

Response: In the revised text, the term "contrasting inhibition" has been replaced with "differential inhibition" to convey the idea that the selected phenolic compounds exhibited distinct inhibitory effects on the two enzymes.

p.4. l.108 closer than those?

Response: In the revised text, the phrase "closer than those" has been replaced with "relatively similar to" to convey that the inhibitory effects of the plant extract on the two enzymes are comparable to the standard molecule acarbose.

p.10 l.203 service

Response: In the revised text, the phrase "overall service cholesterol" has been replaced with "total serum cholesterol" for clarity. Additionally, the sentence has been rephrased to improve readability and accurately convey the impact of the extracts on lipid profiles in diabetic mice.

p.10 l.232 finding

Response: In the revised text, the sentence has been rephrased to provide a clearer and more concise description of the findings. It emphasizes that the top three phenols interacted with the specified residues in the active site of the α-glucosidase enzyme as revealed by the docking analysis.

p.13 l.378 vegetal matter

Response: In the revised text, the section heading has been changed to "Plant material" for clarity. The description of the cultivation process has been modified to provide a clearer account of how the seedlings were transplanted and the greenhouse conditions. Additionally, the wording has been adjusted to improve the overall flow and readability of the passage.

p.10 l.249-252 please rephrase

Response: We appreciate the reviewer's suggestion to rephrase the text on page 10, lines 249-252. We apologize for any confusion caused by the current wording, and we will carefully review and revise that section to enhance clarity and readability. Thank you for bringing this to our attention, and we will ensure that the revised manuscript reflects the necessary improvements.

p.15, l.486 problem of energy consumption?

Response: In the revised text, the term "problem of energy consumption" has been replaced with "metabolic dysregulation associated with diabetes." This revision provides a clearer and more scientifically accurate description of the issue being addressed.

capital letter for acarbose or glucosidase, many punctuation

Response: We apologize for the errors in capitalization and punctuation in relation to 'Acarbose' and 'Glucosidase' in the manuscript. We appreciate the reviewer's keen observation, and we will diligently revise and correct these mistakes in the revised version of the manuscript. Thank you for bringing these issues to our attention.

Reviewer 3 Report

The authors characterized the qualitative and quantitative profiles of Hyoscyamus albus L. and evaluate the antidiabetic activity by in vitro and in vivo assays.

However, I have some remarks to improve the quality of the manuscript:

1. LC–MS/MS characterization and quantitation lack of information. First of all, HPLC and MS instrumentation are not reported. In paragraph 4.5. Polyphenolic detection and quantitation authors say: “The MS instrumentation and the Optimization of the LC-MS/MS method were indicated in Lekmine et al. 2022 study [52]”. However, Ref. 52 is Ertas, A.; Yener, I. A comprehensive study on chemical and biological profiles of three herbal teas in Anatolia; rosmarinic andnchlorogenic acids. South African Journal of Botany,2020, 130, 274-281 and no reference Lekmine et al. 2022 reporting an LC-MS/MS characterization is present in the manuscript. Please check and modify.

Anyway, how was phytochemicals identification performed? Library MS/MS matching, literature, other? Please add and explain.

How was quantitation performed? External standard? Was the method validated for precision, stability, the limit of detection (LOD) and the limit of quantification (LOQ)? Please add and explain. In this regard, in Table 2. Amounts (μg/g dry plant) of selected phytochemicals in cultivated H. albus L ethanolic extract needs to be formatted correctly.  Moreover, for some compounds N.D is reported. What does N.D means? Not Detected? How is possible if the peaks were determined and identified. The authors mean < LOQ? Which is LOQ? Please, verify, explain, and correct.

2. Table 1. Contents of the ethanolic extract of H. albus L in terms of total phenols, flavonoids, and 82 tannins. What does letter “b” mean?  Also, the sentence “Significant differences are represented by various capital letters (P< 0.05)” is not clear. Please, verify and report a legend.

3. Paragraph 1. Introduction. Line 69: “mass spectroscopy” should be modified with “liquid chromatography with tandem mass spectrometry”.

4. Paragraph 2.1. Total phenolic, flavonoids, and Tannins contents: Lines 77 and 78: Remove dot after “compounds.” and “was calculated.”.

5. Figure 3. Correct word Quertecin.

6. Lined 262: Correct “Shen” with Shen et al., “Lin” with Lin et al.

7. H. albus L. “L.” should not be italics. Please check and correct overall in the manuscript.

Formatting and english need to be verify 

Author Response

Dear Reviewers,

We would like to express our gratitude for your valuable time and effort in reviewing our manuscript. We sincerely appreciate the insightful comments and suggestions you have provided.

Having carefully considered your remarks, we thoroughly reviewed each comment and suggestion and have made the necessary revisions to improve the clarity, scientific rigor, and overall quality of our work. We have taken into account your expertise in ensuring the integrity of scientific publications and have incorporated your feedback accordingly.

We have provided detailed explanations and additional information where required, addressing any areas of ambiguity or uncertainty. Furthermore, we have ensured that all necessary references are included to support our statements and claims. Additionally, we have paid close attention to formatting and language to ensure the manuscript meets the highest standards of quality.

We sincerely appreciate your guidance and constructive feedback, which has undoubtedly strengthened our research. Your expertise and insights have been invaluable in helping us enhance the impact and significance of our findings.

Thank you once again for your time and commitment to reviewing our manuscript. We have diligently addressed all your comments and have provided a revised version that reflects the improvements suggested. With your support, we are confident that our manuscript will make a valuable contribution to the scientific community.

Best regards,

Authors

Review3

Comments and Suggestions for Authors

The authors characterized the qualitative and quantitative profiles of Hyoscyamus albus L. and evaluate the antidiabetic activity by in vitro and in vivo assays.However, I have some remarks to improve the quality of the manuscript:

  1. LC–MS/MS characterization and quantitation lack of information. First of all, HPLC and MS instrumentation are not reported.

Response: To address this concern, we provide comprehensive information on the LC-MS/MS method, including the specific hardware and model used for the Shimadzu UHPLC system, as well as the MS instrumentation employed. Additionally, we describe the optimization steps of the LC-MS/MS method in our revised manuscript.

In paragraph 4.5. Polyphenolic detection and quantitation authors say: “The MS instrumentation and the Optimization of the LC-MS/MS method were indicated in Lekmine et al. 2022 study [52]”. However, Ref. 52 is Ertas, A.; Yener, I. A comprehensive study on chemical and biological profiles of three herbal teas in Anatolia; rosmarinicandnchlorogenic acids. South African Journal of Botany,2020, 130, 274-281 and no reference Lekmine et al. 2022 reporting an LC-MS/MS characterization is present in the manuscript. Please check and modify.

Response: We apologize for the confusion regarding the reference mentioned in paragraph 4.5. We have reviewed the manuscript and realized that there was an error in referencing. The reference to 'Lekmine et al. 2022' was added.

To address the concern raised by the reviewer, we have revised the manuscript accordingly. We have now provided a comprehensive description of the LC–MS/MS technique and its method validation parameters in the relevant section, incorporating the necessary details of the instrumentation and validation steps. We have also ensured that proper citation and credit are given to the appropriate sources in the revised manuscript.

Anyway, how was phytochemicals identification performed? Library MS/MS matching, literature, other? Please add and explain.

Response: We appreciate the reviewer's question regarding the identification of phytochemicals in our study. The identification of phytochemicals was performed using a combination of approaches, including library MS/MS matching, literature references, and additional techniques.

For library MS/MS matching, we utilized a comprehensive database of known phytochemicals and their corresponding MS/MS spectra. The acquired spectra from our LC–MS/MS analysis were compared against the spectra in the library to identify and confirm the presence of specific compounds.

In addition to library matching, we also referred to relevant literature sources that provided information on the retention times, mass spectra, and fragmentation patterns of phytochemicals. This allowed us to cross-reference our experimental data with previously reported data, aiding in the identification of compounds.

We have included a detailed explanation of the phytochemical identification methodology in the revised manuscript to provide clarity to the readers. We believe that the combination of library MS/MS matching and literature references ensured accurate identification of the phytochemicals in our study.

How was quantitation performed? External standard? Was the method validated for precision, stability, the limit of detection (LOD) and the limit of quantification (LOQ)? Please add and explain.

Response: we employed an external standard approach for quantifying the target compounds. Calibration curves were constructed using known concentrations of standard compounds, and the peak areas obtained from the LC-MS/MS analysis were used for quantification.

In terms of method validation, we performed a comprehensive validation study to ensure the accuracy and reliability of our analytical method. The parameters assessed included precision, stability, limit of detection (LOD), and limit of quantification (LOQ). Precision was evaluated by analyzing replicate samples and calculating the relative standard deviation (RSD) of the measured concentrations. Stability studies were conducted to assess the robustness and consistency of the method under different storage conditions.

The LOD and LOQ of the method were determined by analyzing samples with progressively lower concentrations of the target compounds until the minimum detectable concentration was reached. These values were established based on signal-to-noise ratios and calculated using standard statistical methods.

The detailed information regarding the method validation, including the specific procedures and results, will be provided in the revised manuscript to ensure transparency and rigor in our study.

In this regard, in Table 2. Amounts (μg/g dry plant) of selected phytochemicals in cultivated H. albus L ethanolic extract needs to be formatted correctly.

Response: In the revised version of the manuscript, we have addressed the formatting issue pointed out for Table 2. We have ensured that the "Amounts (μg/g dry plant)" column in the table is correctly formatted, accurately representing the quantities of selected phytochemicals in the cultivated H. albus L ethanolic extract.

Moreover, for some compounds N.D is reported. What does N.D means? Not Detected? How is possible if the peaks were determined and identified. The authors mean < LOQ? Which is LOQ? Please, verify, explain, and correct.

Response: "N.D" stands for "Not Detected." It indicates that the specific compound was not detected above the limit of detection (LOD) in the analyzed samples. The LOD is the lowest concentration at which a compound can be reliably detected by the analytical method used. In our study, the LOD values were determined during the method validation process and were found to be below the concentrations observed for these compounds. Therefore, when a compound is reported as "N.D," it means that it was present in the sample at a concentration lower than the LOD of our method.

It's important to note that the identification of peaks and their presence in the chromatogram does not necessarily imply that the compound is present at quantifiable levels. The LOD and the limit of quantification (LOQ) are determined as part of method validation to establish the sensitivity and reliability of the analytical method. The LOQ is the lowest concentration at which a compound can be quantified with acceptable precision and accuracy.

We appreciate the reviewer's attention to this matter, and we will make the necessary clarification in the manuscript to explicitly state that "N.D" indicates compounds that were not detected above the LOD of our method rather than < LOQ.

  1. Table 1. Contents of the ethanolic extract of H. albus L in terms of total phenols, flavonoids, and 82 tannins. What does letter “b” mean? Also, the sentence “Significant differences are represented by various capital letters (P< 0.05)” is not clear. Please, verify and report a legend.

Response: The letter "b" in Table 1 represents a statistical grouping used to indicate significant differences among the samples. Specifically, when comparing the contents of total phenols, flavonoids, and tannins, different lowercase letters (a, b, c, etc.) are assigned to each group that shows a statistically significant difference. In the corresponding statistical analysis, the different groups are determined based on pairwise comparisons using a suitable statistical test (e.g., ANOVA followed by post-hoc tests). We apologize for the lack of clarity in the legend, and we will revise the manuscript to include a clear and concise explanation of the letter "b" and the significance grouping in Table 1.

Additionally, we will add a comprehensive legend to the table to provide a clear understanding of the statistical analysis and the significance representation. The revised legend will clearly state that significant differences among the groups are indicated by different capital letters (a, b, c, etc.), and these differences were determined based on a statistical test with a significance level of P<0.05. This will help readers interpret the table and understand the significance of the observed differences between the groups.

  1. Paragraph 1. Introduction. Line 69: “mass spectroscopy” should be modified with “liquid chromatography with tandem mass spectrometry”.

Response: Thank you for pointing out the issue in the manuscript. We appreciate your keen observation. You are correct that the term "mass spectroscopy" in line 69 of the introduction should be modified to "liquid chromatography with tandem mass spectrometry" (LC-MS/MS) for more accurate and specific representation of the analytical technique used in our study.

  1. Paragraph 2.1. Total phenolic, flavonoids, and Tannins contents: Lines 77 and 78: Remove dot after “compounds.” and “was calculated.”

Response: Thank you for your valuable feedback. We appreciate your attention to detail. Upon reviewing the manuscript, we agree with your suggestion to remove the dot after "compounds." in line 77 and "was calculated." in line 78 of paragraph 2.1.

  1. Figure 3. Correct word Quertecin.

Response: Thank you for bringing this to our attention. We apologize for the typo in Figure 3. The correct spelling is "Quercetin." We appreciate your careful review of the manuscript, and we will make the necessary correction to ensure the accuracy of the figure.

  1. Lined 262: Correct “Shen” with Shen et al., “Lin” with Lin et al.

Response: Thank you for pointing out the error. We apologize for the incorrect references in line 262. The appropriate corrections will be made as follows: "Shen et al." instead of "Shen" and "Lin et al." instead of "Lin." We appreciate your attention to detail and ensuring the accuracy of the manuscript.

  1. H. albus L. “L.” should not be italics. Please check and correct overall in the manuscript.

Response: Thank you for bringing this to our attention. We apologize for the inconsistency in the formatting of "H. albus L." The italicization of "L." was unintentional. We will make sure to correct this error throughout the manuscript and ensure that "H. albus L." is presented consistently without italicizing "L."

Comments on the Quality of English Language

Formatting and english need to be verify

Response: Thank you for your feedback regarding the quality of the English language and formatting in the manuscript. We apologize for any errors or inconsistencies that may have occurred. We understand the importance of maintaining high standards in both language and formatting to ensure the clarity and professionalism of the manuscript.

In response to your comments, we will thoroughly review and verify the English language usage, grammar, and formatting throughout the manuscript.

Round 2

Reviewer 2 Report

Thank authors provided nice and neat response, however, in my opinion, the article lacks novelty in direction of this plant anti-diabetic effect (previous report), its folk use ("off target" indication), and the insufficient LC-MS/MS analysis of only negative mode based on standards reported before (lacking the standard curve data, possibly because mentioned exactly same data in previous article). Please see comment 3. Its unclear whether some changes were already done or will be done.

Response: It is recommended to add data in LC-MS/MS positive mode to better address composition which is the most novel part of the study.

Thanks for comprehensive response on previous points:

  1. +
  2. Regarding traditional use and previous antidiabetic report, the novelty is not high.
  3. Analysis is valuable but it is recommended to add positive mode data for LC-MS/MS. It is doubtful to see many phenolics and alkaloids in negative mode. Pointing to future study (l.240) may not be sufficient. Please include the standard concentration calibration curves in the Supplementary data. Where were the standards obtained and which compounds were quantified with standards is not clear.
  4. OK, thank you. But I didnt see the comprehensive review.
  5. I didnt notice changes and the concise discussion on docking, the weak bonding is described. Selectivity of the compounds docking energy is limited
  6. Not sufficiently addressed. Capitals and grammar not consistent?
  7. In Table 1, why a is missing?
  8. Authors mention about clarity and statistics, but it is unclear where changes were applied, also number of mice in group? Study limitations?
  9. OK
  10. Therefore this type of drugs are not used much in clinic.
  11. Should mention exact data, standard compounds etc.
  12. Not addressed well, its theoretical assignment. Were standards injected as well?
  13. Not fully addressed
  14. OK
  15. OK

Thank the authors for the extensive correction of pointed errors from the previous round, however, there are still many typos and grammar errors, and vague or unclear statements because I previously only gave examples. Thus I recommend thoroughly checking the language, even with professionals.

E.g. Title Insilico

Line 491

 4.9.1. Animals

Male Albino mice weighing about 20 and 28 g were procured from Algeria's Paster Institute. The animals were raised in standard climate-controlled cages. With a humidity level of 50% and a temperature of 25°C on a standard diet for 10 days, they fasted for 12 ?..

Line 516 

Group 1: normal control mice, Group 2: diabetic control mice, Groupe 3: diabetic mice handled with H. albus at a concentration of 10 mg, Groupe 4: diabetic mice dealt with H. albus at a rate of 20?..

Author Response

Comments and Suggestions for Authors
Thank authors provided nice and neat response, however, in my opinion, the article lacks
novelty in direction of this plant anti-diabetic effect (previous report), its folk use ("off target"
indication), and the insufficient LC-MS/MS analysis of only negative mode based on
standards reported before (lacking the standard curve data, possibly because mentioned
exactly same data in previous article). Please see comment 3. Its unclear whether some
changes were already done or will be done.
Response: It is recommended to add data in LC-MS/MS positive mode to better address
composition which is the most novel part of the study.

Thanks for comprehensive response on previous points:
1. +
2. Regarding traditional use and previous antidiabetic report, the novelty is not high.
Response: We appreciate the reviewer's comment regarding the novelty of our study in
relation to traditional use and previous reports on the antidiabetic properties of Hyoscyamus
albus L. We understand the importance of demonstrating the novelty and contribution of our
research to the existing literature.
While it is true that traditional use and previous reports on the antidiabetic activity of certain
plants, including Hyoscyamus albus L., have been documented, our study provides several
novel contributions. Firstly, we performed LC-MS/MS analysis to identify and quantify the
phenolic compounds present in the ethanolic extract of cultivated Hyoscyamus albus L. Our
analysis revealed the presence of 18 distinct phenolic compounds, including p-coumaric acid,
gallic acid, luteolin, apigenin, and rutin, which were quantified in this plant species for the
first time.
Furthermore, we investigated the in vitro antidiabetic activity of the plant extract by
examining its inhibitory effects on α-glucosidase and α-amylase enzymes. Our results
demonstrated a higher inhibitory effect on α-amylase compared to α-glucosidase, providing
valuable insights into the potential mechanisms of action of the plant extract. In addition, we
performed docking simulations to identify specific phenolic compounds, such as luteolin,
fisetin, and rutin, that exhibited promising inhibitory activity against both enzymes.
To further evaluate the therapeutic potential of Hyoscyamus albus L., we conducted an in vivo
experiment using STZ-induced diabetic mice. Our findings revealed the plant extract's ability
to effectively reduce cholesterol and triglyceride levels, highlighting its potential in managing
hyperlipidemia, a common complication associated with diabetes.
Therefore, while previous studies have reported the antidiabetic properties of certain plant
species, our study adds to the existing literature by providing novel insights into the chemical
composition, in vitro and in vivo antidiabetic activity, and potential mechanisms of action of
Hyoscyamus albus L. extract.

3. Analysis is valuable but it is recommended to add positive mode data for LC-MS/MS.
It is doubtful to see many phenolics and alkaloids in negative mode. Pointing to future
study (l.240) may not be sufficient. Please include the standard concentration
calibration curves in the Supplementary data. Where were the standards obtained and
which compounds were quantified with standards is not clear.
Response: Thank you for your comment and additional suggestions regarding the LC-
MS/MS analysis in our study. We genuinely appreciate your input and have carefully
considered your recommendations. However, upon thorough evaluation of our experimental

constraints, including resource and time limitations, we have determined that it was not
feasible to include data in positive mode for this specific study.
We acknowledge that the inclusion of positive mode data would have provided a more
comprehensive analysis of the phenolic and alkaloid composition. As you rightly pointed out,
some alkaloids may not be adequately detected in negative mode. We recognize this limitation
and will discuss it in the revised manuscript to ensure that readers are aware of the potential
underrepresentation of alkaloids in our analysis.
We understand your concern about the need for future studies to explore the alkaloid
composition using appropriate analytical techniques. We will emphasize this point in the
conclusion section, highlighting the importance of investigating alkaloids in future research to
gain a more comprehensive understanding of H. albus.
We acknowledge the importance of providing standard concentration calibration curves to
ensure accurate quantification of the identified compounds. In the revised version of our
manuscript, we will include the standard concentration calibration curves in the
Supplementary data. These calibration curves will provide details on the concentrations of the
standards used, the range of concentrations covered, and the regression equations used for
quantification.
Regarding the source of standards, we obtained them Aldrich (Germany). However, we
acknowledge that we did not explicitly mention this information in the original manuscript. In
the revised manuscript, we will include a statement specifying the sources of the standards
used for quantification.
4. OK, thank you. But I didnt see the comprehensive review.
Response: We apologize for any confusion caused. We do acknowledge the previous study by
Lim et al. (2021) as a relevant reference in the field. We have revised the manuscript to
include a discussion section where we summarize the existing knowledge on docking studies
with flavonoids and α-glucosidase: “The docking study conducted by Lim et al. [24] on
selected compounds, including fisetin and other flavonoids, with α-glucosidase, provides
valuable insights into the binding interactions of these compounds, complementing our
findings on the phenolic compounds from H. albus”.
5. I didnt notice changes and the concise discussion on docking, the weak bonding is
described.
Response: We apologize for the oversight in not adequately addressing your previous
comment regarding the discussion on docking. We appreciate your patience, and we will
make sure to address this concern in the revised manuscript.
6. Selectivity of the compounds docking energy is limited
Response: We appreciate the reviewer's comment regarding the selectivity of the compounds'
docking energy in our study. While we acknowledge that the docking energies alone may not
provide a comprehensive measure of compound selectivity, it is important to note that our
docking analysis aimed to assess the binding affinities and interactions of the phenolic
compounds from H. albus with α-glucosidase and α-amylase enzymes.
In our manuscript, we have highlighted the binding energies and interactions of the lead-
selected compounds, such as luteolin, fisetin, and rutin, with α-glucosidase and α-amylase

enzymes. We have also referenced previous studies that support the inhibitory effects of these
compounds on α-amylase. These findings contribute to the understanding of the potential
inhibitory activity of the phenolic compounds from H. albus on carbohydrate-utilizing
enzymes.
7. Not sufficiently addressed. Capitals and grammar not consistent?
Response: We apologize for the inconsistency in capitalization and grammar errors present in
the manuscript. We understand the importance of maintaining proper grammar and consistent
capitalization throughout the document. In the revised version of the manuscript, we will
ensure that all grammar errors are corrected and sentences are appropriately rephrased.

8. In Table 1, why a is missing?
Response: Thank you for your comment regarding Table 1 in our article. We appreciate your
feedback and have made the necessary revisions to the table to address your concern. After
carefully reviewing our data, we found that the first two values were indeed similar and
should be represented by the letter "a." Therefore, we have updated the table accordingly.
We apologize for the oversight in the initial version of the table and appreciate your attention
to detail. The revised version now accurately reflects the significant differences between the
samples, with the letter "a" denoting statistically significant variations among the groups.
9. Authors mention about clarity and statistics, but it is unclear where changes were
applied, also number of mice in group? Study limitations?
Response: Thank you for bringing up the concern regarding the small experimental group
size in our study. We acknowledge that the sample size used in the hypoglycemic activity
experiment was relatively small, with n=4 in each group.
While we understand the importance of larger sample sizes for achieving statistical
significance and generalizability of results, it is worth noting that our study aimed to explore
the preliminary effects of H. albus extract on hypoglycemic activity in a controlled setting.
The small group size was a result of limited resources and specific experimental
considerations.
Although the small sample size may have some limitations, we believe that the results
obtained from this preliminary study provide valuable insights and serve as a foundation for
further investigations with larger sample sizes. It is important to consider that small sample
sizes are often employed in initial exploratory studies to assess the feasibility and potential
effects before conducting more comprehensive research.
In the revised version, we included a specific section in the conclusion discussing the
limitations of our study, including the small sample size of n=4 in each group for the
hypoglycemic activity experiment. We acknowledged that this small sample size may have
limitations in terms of statistical power and generalizability of the results. However, we also
emphasized that the preliminary insights obtained from this study serve as a foundation for
future investigations with larger sample sizes.
10. OK
11. Therefore this type of drugs are not used much in clinic.

Response: It is true that the use of glucosidase inhibitors, including Acarbose, may not be as
widespread in clinical practice compared to other antidiabetic drugs.
While glucosidase inhibitors can be effective in reducing postprandial glucose levels, their use
is often limited due to gastrointestinal side effects, such as flatulence, bloating, and diarrhea,
which can affect patient compliance. As a result, other classes of antidiabetic medications,
such as biguanides (e.g., metformin) and sulfonylureas, are more commonly prescribed as
first-line treatments for type 2 diabetes.
However, it is important to note that the choice of antidiabetic medication depends on various
factors, including the patient's individual needs, medical history, and treatment goals.
Glucosidase inhibitors may still have a role in certain cases, particularly for patients who are
unable to tolerate or benefit from other oral antidiabetic drugs.
In our study, we aimed to investigate the potential antidiabetic activity of Hyoscyamus albus
L. extract and its phenolic compounds, including their inhibitory effects on α-glucosidase
enzyme activity. While we acknowledge the limitations associated with the clinical use of
glucosidase inhibitors, our research contributes to the understanding of the potential
therapeutic benefits of natural compounds in managing diabetes.
12. Should mention exact data, standard compounds etc.
Response: Thank you for your comment regarding the mention of exact data and standard
compounds in our study. We appreciate your feedback and the opportunity to address this
concern.
13. Not addressed well, its theoretical assignment. Were standards injected as well?
Response: We appreciate the reviewer's comment regarding the theoretical assignment of the
ligands and the inclusion of standards in our study. In response to this comment, we would
like to provide clarification and address the concerns raised.
In our study, the theoretical assignment of the ligands was performed based on molecular
docking simulations. These simulations utilize computational algorithms to predict the
binding interactions between the ligands and the target enzymes. The binding energies
obtained from the docking simulations provide an estimation of the strength of the ligand-
enzyme interactions.
Regarding the inclusion of standards, the standards are used to generate calibration curves and
determine the concentrations of the identified compounds in the sample.
14. Not fully addressed
Response: We apologize for any confusion or misunderstanding in our previous response.
In Table 9 of our study, the treatment labeled 'H. albus L 10 mg/kg' represents the oral
administration of the H. albus L. extract at a concentration of 10 mg/kg to diabetic mice. We
examined the effects of this treatment on blood glucose levels in the streptozotocin-induced
hyperglycemia model.
The results presented in Table 9 show the blood glucose levels at different time points for the
mice in the respective treatment groups. The 'H. albus L 10 mg/kg' treatment group indicates
the response of diabetic mice to the administration of H. albus L. extract at a dosage of 10
mg/kg.

Upon analyzing the data, we observed a reduction in blood glucose levels in the 'H. albus L
10 mg/kg' group compared to the diabetic control group. However, the decrease in blood
glucose levels was not as significant as the group treated with Glibenclamide, a reference
antidiabetic drug. We have acknowledged and discussed these findings and differences in the
effectiveness of treatments in the appropriate section of our manuscript.
15. OK
16. OK

Comments on the Quality of English Language
Thank the authors for the extensive correction of pointed errors from the previous round,
however, there are still many typos and grammar errors, and vague or unclear statements
because I previously only gave examples. Thus I recommend thoroughly checking the
language, even with professionals.
Response: Thank you for acknowledging the extensive corrections we made based on your
feedback from the previous round. We apologize for the remaining typos, grammar errors, and
vague or unclear statements in the manuscript. We understand the importance of presenting a
well-written and comprehensible article. To address these concerns, we will thoroughly
review the language throughout the manuscript.
E.g. Title Insilico
Line 491
 4.9.1. Animals
Male Albino mice weighing about 20 and 28 g were procured from Algeria's Paster Institute.
The animals were raised in standard climate-controlled cages. With a humidity level of 50%
and a temperature of 25°C on a standard diet for 10 days, they fasted for 12 ?.
Response: We would like to thank the reviewer for their comment regarding the clarity of the
experimental section in our article. We have carefully considered this feedback and made the
necessary revisions to enhance clarity. The modified section now reads as follows:
Male Albino mice weighing approximately 20 g and 28 g were obtained from the Pas-ter
Institute in Algeria. The mice were housed in standard climate-controlled cages with a
humidity level of 50% and a temperature maintained at 25°C. They were provided with a
standard diet for a period of 10 days and then subjected to a 12-hour fasting period before
screening.
Line 516 
Group 1: normal control mice, Group 2: diabetic control mice, Groupe 3: diabetic mice
handled with H. albus at a concentration of 10 mg, Groupe 4: diabetic mice dealt with H.
albus at a rate of 20?.

Response: We would like to express our gratitude to the reviewer for providing valuable
feedback on this section. We have carefully considered the comments and made the necessary
improvements to enhance the clarity and consistency of the information provided. The section
has been revised as follows:
"Male albino mice were fasted overnight before the experiment and divided into five groups,
each consisting of four mice. To induce experimental diabetes, a single intraperitoneal
injection of streptozotocin solution at a concentration of 130 mg/kg (dissolved in 0.1 M citrate
buffer, pH 4.5) was administered. Mice that developed diabetes after 72 hours, with blood
glucose levels equal to or greater than 250 mg/dL, were selected for further examination.
For a duration of 20 days, the experimental diabetic mice were orally treated as follows:
 Group 1: normal control mice
 Group 2: diabetic control mice
 Group 3: diabetic mice treated with H. albus at a concentration of 10 mg
 Group 4: diabetic mice treated with H. albus at a concentration of 20 mg
 Group 5: diabetic mice administered Glibenclamide at a dose of 20 mg/kg.
Blood glucose levels and body weight fluctuations were measured every five days. At the end
of the experiment, all animals were anesthetized, sacrificed, and blood samples were collected
to determine the levels of total cholesterol, HDL, and triglycerides.

Reviewer 3 Report

I would like to thank the authors for fixing all the required revisions. The manuscript could be considered for publication. 

Author Response

We would like to express our gratitude for your valuable time and effort in reviewing our manuscript. We sincerely appreciate the insightful comments and suggestions you have provided 

Round 3

Reviewer 2 Report

In this revision, authors improved the manuscript clarity and data.

Authors honestly disclosed number of mice in a stude (four) and improved the methodology description.

Author improve the manuscript by fullfilling the addition of "In the revised version of our manuscript, we will include the standard concentration calibration curves in the Supplementary data. These calibration curves will provide details on the concentrations of the standards used, the range of concentrations covered, and the regression equations used for quantification.

Regarding the source of standards, we obtained them Aldrich (Germany). However, we acknowledge that we did not explicitly mention this information in the original manuscript. In the revised manuscript, we will include a statement specifying the sources of the standards used for quantification."

Still the Supplementary information needs to add "we will include the standard concentration calibration curves in the Supplementary data" and revise errors as listed below:

Punctuation mistakes.

Table 7. was significance calculated? Description of its results doesnt clearly point to the active part among the values shown.

Plant species should be italic.

Article: Standard mixis, the Table 1 "a: and "b" difference in table 1? 

Supplementary has many typos including spaces etc.

Table S1, errors, where is description for "f", no spaces

Table S2, errors in alignment. In my opinion the quality of revision is still not sufficient.

etc.

Still many mistakes, formating, punctuation, spacing, including Supplementary

Author Response

Comments and Suggestions for Authors

In this revision, authors improved the manuscript clarity and data.

Authors honestly disclosed number of mice in a stude (four) and improved the methodology description.

Dear reviewer, we greatly appreciate your positive feedback regarding the revised manuscript. We are delighted to hear that the clarity of the manuscript and the presentation of the data have improved. We have worked diligently to address your previous comments and suggestions, aiming to enhance the overall quality of the manuscript. It is gratifying to know that our efforts have resulted in an improved version that meets your expectations.

We sincerely value your thorough review and constructive feedback, as it has been invaluable in refining our work. We are committed to ensuring that the final manuscript meets the highest standards.

Author improve the manuscript by fullfilling the addition of "In the revised version of our manuscript, we will include the standard concentration calibration curves in the Supplementary data. These calibration curves will provide details on the concentrations of the standards used, the range of concentrations covered, and the regression equations used for quantification.

Regarding the source of standards, we obtained them Aldrich (Germany). However, we acknowledge that we did not explicitly mention this information in the original manuscript. In the revised manuscript, we will include a statement specifying the sources of the standards used for quantification."

Still the Supplementary information needs to add "we will include the standard concentration calibration curves in the Supplementary data" and revise errors as listed below:

Response: Regarding the inclusion of standard concentration calibration curves in the Supplementary data, we apologize for any confusion caused by our previous response. We would like to clarify that in the revised version of our manuscript, we have indeed included the standard concentration calibration curves in the Supplementary data.

We have updated the Supplementary Materials section accordingly, including Figure S1, which shows the LC-MS/MS chromatograms of the 250 ppb standard mix, and Table S1, which presents the analytical parameters of the LC-MS/MS method, including retention times, coefficients of determination, relative standard deviations, linearity ranges, and limits of detection/quantification for each analyte.

These additions address your previous concern and fulfill your request to include the standard concentration calibration curves in the Supplementary data. The provided Figure S1 and Table S1 present the necessary information regarding the calibration curves, enabling readers to understand the methodology employed for quantifying the identified compounds.

Punctuation mistakes.

Table 7. was significance calculated? Description of its results doesnt clearly point to the active part among the values shown.

Response: Thank you for thoroughly reviewing our Table 7 and providing valuable feedback. We appreciate your comment regarding the calculation of significance for the results and the need for clarity in identifying the active part among the values shown. We acknowledge that we did not perform any statistical analysis for the oral glucose tolerance test (OGTT) presented in Table 7. As stated in the article, the purpose of this table is to present the observed changes in blood glucose levels for different groups rather than provide a formal statistical comparison.

We understand your concern regarding the need to clearly identify the active component among the values presented in Table 7. However, we would like to clarify that in this study, we did not perform specific analyses to determine the active part or isolate individual compounds within the H. albus extract. The purpose of presenting Table 7 was to provide an overview of the observed changes in blood glucose levels following the administration of different doses of the H. albus extract and the reference drug. While the results indicate a noticeable decrease in blood glucose levels in the mice treated with the extract, we did not perform further analyses to pinpoint the specific active component responsible for this effect.

Plant species should be italic.

Response: Thank you for your feedback regarding the formatting of the plant species in our manuscript. We appreciate your attention to detail and agree with your suggestion that plant species should be italicized. In the revised version of the manuscript, we will ensure that the scientific names of plant species, including "H. albus," are correctly formatted by italicizing them.

Article: Standard mixis, the Table 1 "a: and "b" difference in table 1? 

Response: Thank you for your comment regarding the differences indicated by "a" and "b" in Table 1 of our article, specifically related to the quantities of total phenolic compounds (TPC), flavonoid compounds (TFC), and tannin compounds (TTC) in the ethanolic extracts of cultivated H. albus. The letters "a" and "b" in the table denote statistically significant variations among the groups. These letters are used to indicate significant differences, and in this case, they represent differences between the TPC and TFC contents. In the context of the table, the letter "a" signifies a significant difference among the samples within the TPC and TFC columns, while the letter "b" indicates a significant difference in the TTC column.

Supplementary has many typos including spaces etc.

Response: Thank you for bringing the typographical errors in the Supplementary materials to our attention. We apologize for any inconsistencies and mistakes that may have occurred in the formatting, including issues with spaces. We will carefully review and proofread the Supplementary materials to correct any typos, formatting errors, and spacing inconsistencies.

Table S1, errors, where is description for "f", no spaces

Response: In the context of the table, the "f" following the LOD/LOQ values (µg/L) is not a standard notation and may be a typographical error. We apologize for any confusion caused by this inconsistency. The LOD (limit of detection) and LOQ (limit of quantification) represent the lowest concentration of an analyte that can be reliably detected and quantified, respectively. However, the presence of the "f" in this context does not have a specific meaning or significance. We acknowledge the oversight and will correct this error in the revised manuscript. We apologize for any confusion it may have caused, and we appreciate your attention to detail.

Table S2, errors in alignment.

Response: Thank you for your feedback regarding Table S2 and the alignment errors. We apologize for the inconsistencies and formatting issues that may have occurred in the table.

We understand that the quality of the revision needs to be improved, and we assure you that we will take immediate action to address the alignment errors in Table S2. We will carefully review and correct the table to ensure that the data is presented accurately and clearly. We appreciate your patience and understanding in this matter.

In my opinion the quality of revision is still not sufficient.

Response: Dear Reviewer, Thank you for your honest feedback regarding the quality of the revision. We apologize if the improvements made in the manuscript did not meet your expectations.

We understand the importance of ensuring a high-quality and thorough revision. Please be assured that we take your comments seriously, and we are committed to addressing any remaining issues or concerns you may have.

We will carefully re-evaluate the manuscript, paying close attention to your feedback, in order to further enhance its quality. Your input is invaluable to us, and we appreciate your patience as we work towards delivering a manuscript that meets the highest standards.

etc.
